# Scaling Beyond Masked Diffusion Language Models

**Subham Sekhar Sahoo** [1 2] **Jean-Marie Lemercier** [† 1] **Zhihan Yang** [† 3] **Justin Deschenaux** [† 4] **Jingyu Liu** [1 5]
**John Thickstun** [3] **Ante Jukić** [1]

## Abstract

Diffusion language models are a promising alternative to autoregressive models due to their potential for faster generation. Among discrete diffusion approaches, Masked diffusion currently dominates, largely driven by strong perplexity on language modeling benchmarks. In this work, **we present the first scaling law study of uniform-state and interpolating discrete diffusion methods**. We also show that Masked diffusion models can be made approximately 12% more FLOPs-efficient when trained with a simple cross-entropy objective. We find that perplexity is informative within a diffusion family but can be misleading across families, where models with worse likelihood scaling may be preferable due to faster and more practical sampling, as reflected by the speed-quality Pareto frontier. **These results challenge the view that Masked diffusion is categorically the future of diffusion language modeling** and that perplexity alone suffices for cross-algorithm comparison. Scaling all methods to 1.7B parameters, we show that uniform-state diffusion remains competitive on likelihood-based benchmarks and outperforms autoregressive and Masked diffusion models on GSM8K, despite worse validation perplexity. We provide the code, model checkpoints, and video tutorials on the project page:

https://s-sahoo.com/scaling-dllms

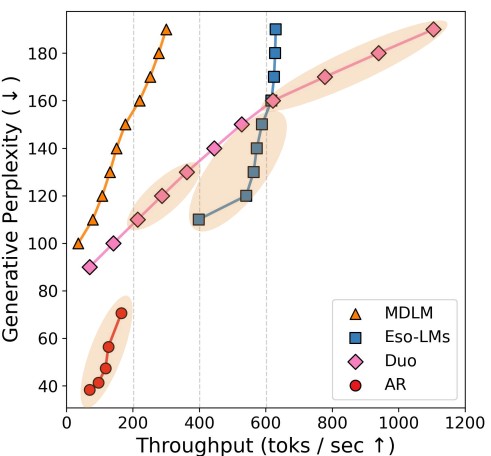

*Figure 1.* **Speed-Quality Pareto Frontier**. We report the highest throughput achieved by compute-optimal models across a range of training FLOPs budgets. AR produces the highest-quality samples but is slow. Sample diversity (measured by entropy) remains broadly similar across algorithms, with Duo exhibiting slightly reduced diversity; see Fig. 5. Duo dominates in the throughput ranges $[200, 400] \cup [600, \infty]$, while Eso-LM dominates in the range $[400, 600]$.

## 1. Introduction

Autoregressive (AR) language models have dominated text generation due to strong likelihoods and a mature training and inference ecosystem. Recently, diffusion large language models (d-LLMs) have emerged as credible alternatives (Song et al., 2025; Labs et al., 2025). Crucially, unlike AR models that decode left-to-right, diffusion models generate by iteratively refining an entire sequence in parallel, which supports a favorable speed-quality trade-off and can enable faster decoding than token-by-token generation. Discrete diffusion models, especially Masked Diffusion Language Models (MDLM), have closed much of the perplexity gap to AR models at small scales (Sahoo et al., 2024a; Shi et al., 2025; Ou et al., 2025; Arriola et al., 2025). At larger scales, e.g., 8B parameters, MDLM can match strong AR baselines on challenging math and science datasets, while also mitigating AR-specific failure modes such as the reversal curse (Nie et al., 2025b). Overall, these findings position d-LLMs as a distinct and competitive paradigm, offering complementary advantages over AR models, particularly parallel decoding and flexible inference-time compute.

---

[†]Joint second authors [1]NVIDIA, Santa Clara, USA [2]Department of Computer Science, Cornell Tech, NY, USA [3]Department of Computer Science, Cornell University, NY, USA [4]School of Computer and Communication Sciences, EPFL Lausanne, Switzerland [5]Department of Computer Science, University of Chicago, Illinois. Correspondence to: Subham Sahoo <ssahoo@cs.cornell.edu>.

*Proceedings of the 43$^{rd}$ International Conference on Machine Learning*, Seoul, South Korea. PMLR 306, 2026. Copyright 2026 by the author(s).

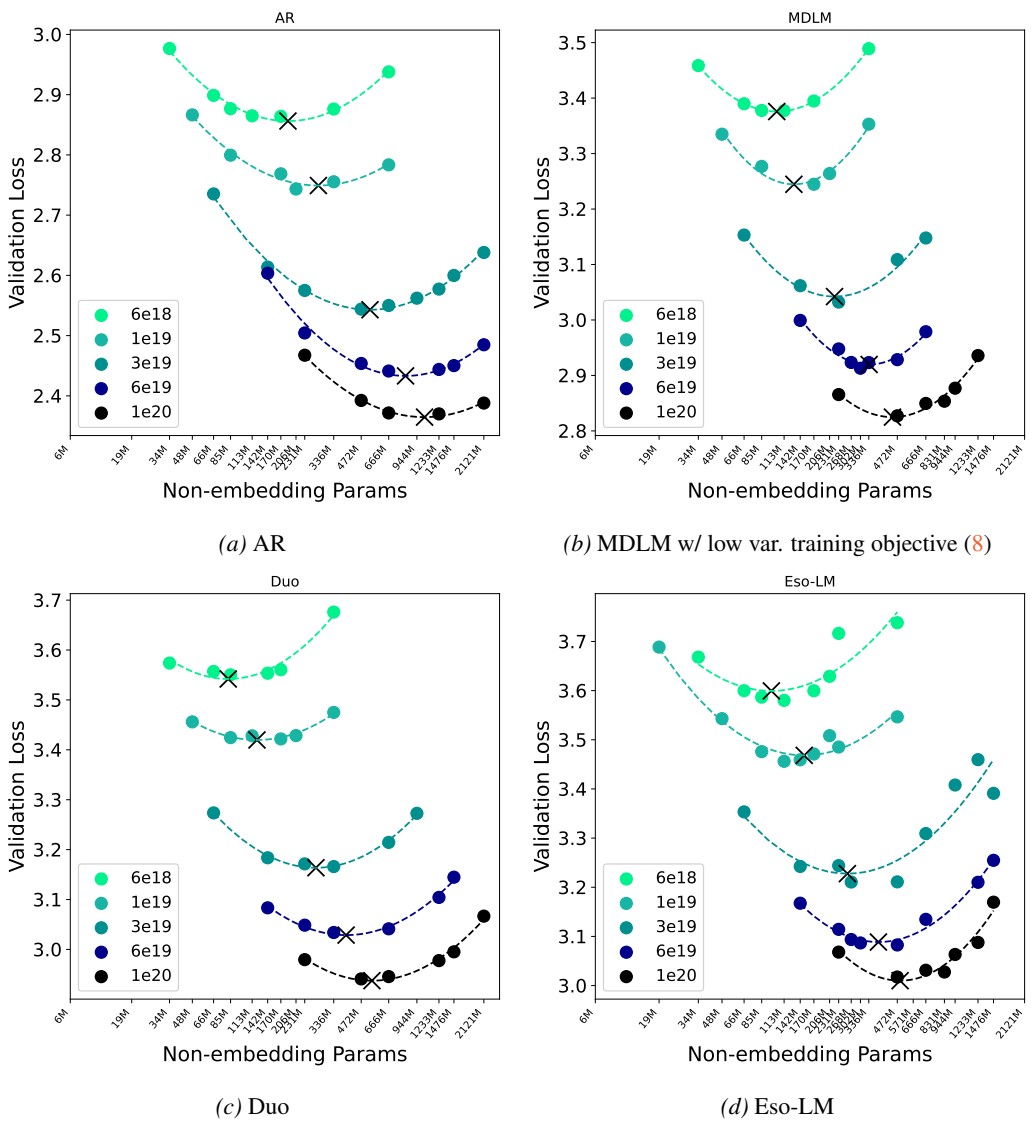

*Figure 2.* **IsoFLOP Analysis** under fixed computation budgets.

D-LLMs are typically optimized and compared using likelihood-based metrics such as validation perplexity. This is natural: perplexity is the canonical language modeling metric and underpins scaling-law analyses that guide compute-optimal allocation between parameters and data (Kaplan et al., 2020; Hoffmann et al., 2022). For d-LLMs, however, this perspective is incomplete. Perplexity does not reflect inference-time behavior. For example, MDLMs can excel at inference-time scaling, where additional sampling compute reliably improves sample quality (Wang et al., 2025), whereas Uniform-state diffusion models (USDMs) can excel in the few-step regime (Sahoo et al., 2025a). Moreover, the true perplexity of d-LLMs is generally intractable, so we rely on bounds. Because different diffusion formulations (as we discuss later) use different

forward noising processes and reverse sampling procedures, they induce different likelihood bounds. Consequently, perplexities across diffusion families aren't comparable; hence, the diffusion family with the best perplexity may not be the most effective in practice. For example, USDMs trail MDLMs in perplexity; however, recent results suggest that improved samplers can substantially change the picture: with stronger sampling procedures, Uniform-state diffusion can outperform Masked diffusion on controllable generation and even on unconditional language generation (Sahoo et al., 2025a; Deschenaux et al., 2026; Schiff et al., 2025). This motivates the question: **Is Masked diffusion the dominant paradigm for non-autoregressive discrete generation, or merely the current front-runner in a larger design space?**

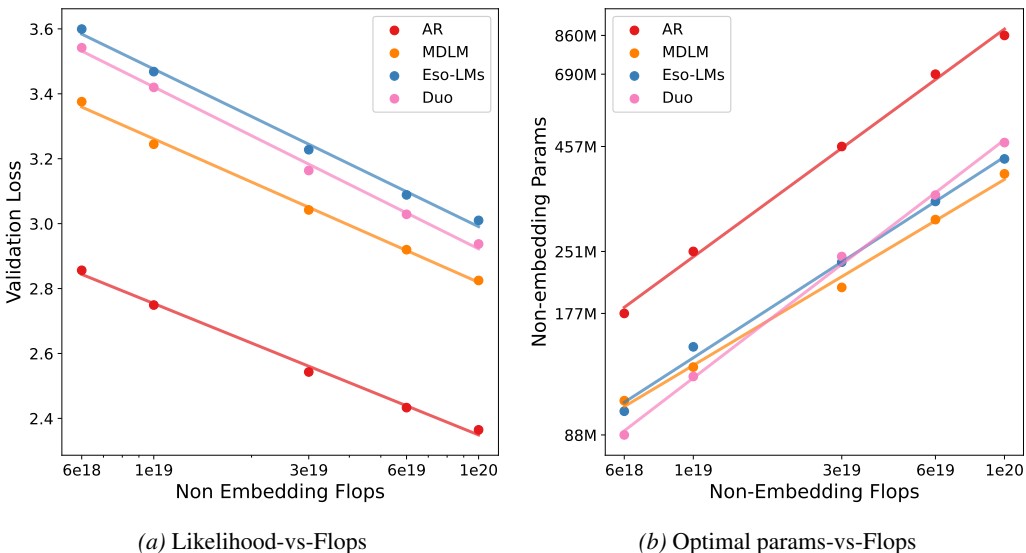

*(a) Likelihood-vs-Flops*    *(b) Optimal params-vs-Flops*

*Figure 3.* **Scaling Laws.** Diffusion models exhibit similar scaling behavior wrt AR models.

In this paper, we address this question through a systematic study of three representative families of discrete diffusion LLMs: Masked diffusion, Uniform-state diffusion, and interpolating diffusion. These families capture distinct strengths. Masked diffusion is the strongest in terms of perplexity (Sahoo et al., 2024a). Uniform-state diffusion, despite worse perplexity, often produces higher-quality samples in the few-step regime (Sahoo et al., 2025a) and is particularly well-suited to guidance (Schiff et al., 2025). Interpolating diffusion methods (Arriola et al., 2025; Sahoo et al., 2025b) support KV caching during inference, enabling substantially faster decoding than other diffusion families. We focus on the state-of-the-art representative from each category: MDLM (Sahoo et al., 2024a) (Masked diffusion), Duo (Sahoo et al., 2025a) (Uniform-state diffusion), and Eso-LM (Sahoo et al., 2025b) (interpolating diffusion).

First, we perform compute-matched scaling studies for all three model families (Sec. 3). Prior work largely focuses on MDLMs (Nie et al., 2025a), leaving other diffusion families underexplored. We fit scaling laws for compute-optimal validation loss and model size, enabling direct comparisons of scaling exponents and constant-factor gaps. Because likelihood alone does not capture inference-time advantages, we also evaluate speed-quality trade-offs by measuring throughput and sample quality across sampling steps, using Gen PPL computed under a strong AR evaluator, and constructing Pareto frontiers (Sec. 3.3; Fig. 1). Finally, we validate these trends at larger scale by training 1.7B-parameter models and evaluating them on likelihood-based benchmarks as well as a math and reasoning dataset (GSM8K; Cobbe et al., 2021).

**Our results challenge the view that Masked diffusion is categorically the future of diffusion language modeling and, more broadly, that perplexity suffices for cross-algorithm comparison.** While MDLM exhibits the strongest likelihood scaling, we show that (i) its scaling can be improved with a low-variance training objective, reducing the compute multiplier to within about 12% of AR while shifting compute-optimal checkpoints toward smaller models, and (ii) diffusion families with worse perplexity scaling, notably Duo and Eso-LM, can dominate the speed-quality Pareto frontier due to more efficient sampling. At 1.7B parameters, these trade-offs persist: Duo remains competitive on likelihood-based downstream evaluations despite worse validation perplexity and, notably, outperforms AR, MDLM, and Eso-LM on GSM8K after supervised fine-tuning (SFT).

Our contributions are as follows:

1. We present the first systematic IsoFLOP scaling study for a state-of-the-art Uniform-state diffusion model (Duo) and an interpolating diffusion model (Eso-LM) (Sec. 3).

2. We show that a low-variance training objective improves MDLM's compute efficiency and shifts compute-optimal checkpoints toward smaller models, which reduces inference cost (Sec. 3.2.2).

3. We demonstrate that perplexity is informative within a family but can be misleading across diffusion families, where models with worse likelihood scaling may be preferable due to faster and more practical sampling, as captured by the speed-quality Pareto frontier (Sec. 3.3; Fig. 1).

4. We scale all methods to 1.7B parameters and show

that uniform-state diffusion remains competitive on likelihood-based benchmarks and achieves the strongest math and reasoning performance after fine-tuning, despite worse validation perplexity (Sec. 4.2).

## 2. Background

**Notation** We denote scalar discrete random variables with $K$ categories as 'one-hot' column vectors and define $\mathcal{V} \in \{\mathbf{v} \in \{0,1\}^K : \sum_{i=1}^K \mathbf{v}_i = 1\}$ as the set of all such vectors. Define $\mathrm{Cat}(\cdot; \boldsymbol{\pi})$ as the categorical distribution over $K$ classes with probabilities given by $\boldsymbol{\pi} \in \Delta^K$, where $\Delta^K$ denotes the $K$-simplex. We also assume that the $K$-th category corresponds to a special [MASK] token and let $\mathbf{m} \in \mathcal{V}$ be the one-hot vector for this mask, i.e., $\mathbf{m}_K = 1$. Additionally, let $\mathbf{1} = \{1\}^K$ and $\langle \mathbf{a}, \mathbf{b} \rangle$ and $\mathbf{a} \odot \mathbf{b}$ respectively denote the dot and Hadamard products between two vectors $\mathbf{a}$ and $\mathbf{b}$. We use $\mathbf{x} \in \mathcal{V}^L$ to represent clean data which is a length-$L$ sequence containing no mask tokens, and let $\mathbf{x}^\ell$ denote its $\ell^{\mathrm{th}}$ element where $\ell$ refers to the token index. Under our notation, each $\mathbf{x}^\ell$ is a one-hot vector.

### 2.1. Autoregressive Models

> **Pro:** Best perplexity among all language models.
> **Con:** Sequential nature prevents parallel generation.

Autoregressive (AR) sequence models define a left-to-right factorization of the data likelihood. For $\mathbf{x} \sim q_{\mathrm{data}}$,

$$\log p_\theta(\mathbf{x}) = \sum_{\ell=1}^L \log p_\theta(\mathbf{x}^\ell \mid \mathbf{x}^{<\ell}), \quad (1)$$

where $p_\theta(\mathbf{x}^\ell \mid \mathbf{x}^{<\ell})$ is typically implemented by a causal Transformer (Vaswani et al., 2017). Generation proceeds sequentially, so producing a length-$L$ sample requires $L$ model evaluations (often counted as NFEs). A practical advantage of causal attention is that one can cache key/value states from earlier positions, substantially reducing the cost of decoding during inference (KV caching).

### 2.2. Discrete Diffusion Models

Discrete diffusion models construct a forward noising process that gradually transforms clean data into a simple prior, and then learn a reverse generative process to map the samples from the prior distribution to data (Sohl-Dickstein et al., 2015; Austin et al., 2021; Campbell et al., 2022). Let $\mathbf{x} \sim q_{\mathrm{data}}$ be a clean sequence and let $\mathbf{z}_t \in \mathcal{V}^L$ denote the latent sequence at time $t \in [0,1]$ produced by the forward process. Typically, the corruption is independent across positions, i.e.,

$$q_t(\mathbf{z}_t \mid \mathbf{x}) = \prod_{\ell=1}^L q_t(\mathbf{z}_t^\ell \mid \mathbf{x}^\ell). \quad (2)$$

In this work we consider "interpolating" forward kernels, where each token marginal is a linear combination of the clean one-hot token and a fixed categorical prior:

$$\mathbf{z}_t^\ell \sim q_t(\cdot \mid \mathbf{x}^\ell; \alpha_t) = \mathrm{Cat}(\cdot; \alpha_t \mathbf{x}^\ell + (1-\alpha_t)\boldsymbol{\pi}). \quad (3)$$

Here $\alpha_t \in [0,1]$ decreases monotonically with $t$ and serves as the noise schedule: $\alpha_0 \approx 1$ corresponds to (nearly) clean data, and $\alpha_1 \approx 0$ corresponds to the prior. The learning objective is to fit a reverse-time model $p_\theta$ parameterized by a neural network with parameters $\theta$ that inverts this corruption, turning samples from the prior back into samples from $q_{\mathrm{data}}$. Training is commonly phrased in terms of a negative variational bound on $\log p_\theta(\mathbf{x})$. A key design choice is the prior $\boldsymbol{\pi}$, which yields two widely used families discussed next.

#### 2.2.1. MASKED DIFFUSION MODELS

> **Pro:** Best perplexity among discrete diffusion models.
> **Con:** Slow sampling due to lack of KV caching.

**Forward process** Masked Diffusion Models (Austin et al., 2021; Lou et al., 2024; Sahoo et al., 2024b; Shi et al., 2025; Ou et al., 2025) instantiate (3) with a mask prior, i.e., $\boldsymbol{\pi} = \mathbf{m}$:

$$q_t(\mathbf{z}_t^\ell \mid \mathbf{x}^\ell) = \mathrm{Cat}(\mathbf{z}_t^\ell; \alpha_t \mathbf{x}^\ell + (1-\alpha_t)\mathbf{m}). \quad (4)$$

Intuitively, as $t$ increases from 0 and 1, the forward process progressively replaces tokens with [MASK] while leaving unmasked tokens unchanged. A common noise schedule is $\alpha_t = 1 - t$, though other monotonically decreasing schedules (e.g., cosine) are also possible.

**Reverse process** For $s < t$, the exact reverse posterior $q_{s|t}(\mathbf{z}_s^\ell \mid \mathbf{z}_t^\ell, \mathbf{x}^\ell)$ has a convenient form due to the absorbing nature of the mask: if $\mathbf{z}_t^\ell \neq \mathbf{m}$, the token must remain fixed; if $\mathbf{z}_t^\ell = \mathbf{m}$, the posterior mixes between the clean token and the mask distribution (Sahoo et al., 2024a):

$$q_{s|t}(\mathbf{z}_s^\ell \mid \mathbf{z}_t^\ell, \mathbf{x}^\ell) = \begin{cases} \mathrm{Cat}(\mathbf{z}_s^\ell; \mathbf{z}_t^\ell) & \mathbf{z}_t^\ell \neq \mathbf{m}, \\ \mathrm{Cat}\left(\mathbf{z}_s^\ell; \frac{(1-\alpha_s)\mathbf{m}+(\alpha_s-\alpha_t)\mathbf{x}^\ell}{1-\alpha_t}\right) & \mathbf{z}_t^\ell = \mathbf{m}. \end{cases} \quad (5)$$

**Training** Let $\mathbf{x}_\theta : \mathcal{V}^L \to (\Delta^K)^L$ be a denoiser (typically a bidirectional Transformer) that outputs a categorical distribution for each position. A standard parameterization of the learned reverse transition replaces the unknown clean token $\mathbf{x}^\ell$ in (5) with a model prediction $\mathbf{x}_\theta^\ell(\mathbf{z}_t)$:

$$p_{s|t}^\theta(\mathbf{z}_s \mid \mathbf{z}_t) = \prod_\ell^L p_{s|t}^\theta(\mathbf{z}_s^\ell \mid \mathbf{z}_t) = \prod_\ell^L q_{s|t}(\mathbf{z}_s^\ell \mid \mathbf{z}_t^\ell, \mathbf{x}^\ell = \mathbf{x}_\theta^\ell(\mathbf{z}_t)). \quad (6)$$

The resulting Negative Evidence Lower Bound (NELBO) is

$$\mathcal{L}_{\mathrm{MDLM}}^{\mathrm{NELBO}}(\mathbf{x}) = \mathbb{E}_{q_t, t \sim [0,1]}\left[\frac{\alpha_t'}{1-\alpha_t} \sum_{\ell \in \mathcal{M}(\mathbf{z}_t)} \log \langle \mathbf{x}_\theta^\ell(\mathbf{z}_t), \mathbf{x}^\ell \rangle\right]. \quad (7)$$

Sahoo et al. (2025b) made a very crucial observation about (7): under the linear schedule $\alpha_t = 1 - t$, the ratio $\alpha_t'/(1-\alpha_t)$ approaches $\infty$ as $t \to 0$. Therefore, Sahoo et al. (2025b) propose to replace this ratio in (7) with $-1$ only during training, which reduces the training variance:

$$\mathcal{L}_{\text{MDLM}}(\mathbf{x}) = -\mathbb{E}_{q_t, t \sim [0,1]} \left[ \sum_{\ell \in \mathcal{M}(\mathbf{z}_t)} \log \langle \mathbf{x}_\theta^\ell(\mathbf{z}_t), \mathbf{x}^\ell \rangle \right]. \quad (8)$$

While using (8) was explored in prior work (Chang et al., 2022; Gat et al., 2024), only Sahoo et al. (2025b) identified the practical benefits of training with this loss.

### 2.2.2. UNIFORM-STATE DIFFUSION MODELS

> **Pro:** Self-correction and few-step sampling.
> **Con:** Worse perplexity than MDLMs.

**Forward Process** Uniform-state Diffusion Models (US-DMs) (Lou et al., 2024; Schiff et al., 2025; Sahoo et al., 2025a) instantiate (3) with a uniform prior $\boldsymbol{\pi} = \mathbf{1}/K$:

$$q_t(\mathbf{z}_t^\ell \mid \mathbf{x}^\ell) = \text{Cat}\Big(\mathbf{z}_t^\ell; \alpha_t \mathbf{x}^\ell + (1 - \alpha_t)\mathbf{1}/K\Big). \quad (9)$$

In contrast to Masked diffusion, the forward process does not involve transitions to the mask token. Instead, it diffuses each token toward the uniform categorical distribution. As a result, the reverse process can revise token values multiple times, enabling "self-correction" and improving few-step sampling and guided generation.

**Reverse Process** The reverse posterior $q_{s|t}^{\text{USDM}}$ also has a closed form:

$$q_{s|t}(\cdot \mid \mathbf{z}_t^\ell, \mathbf{x}^\ell) = \text{Cat} \left( \cdot; \frac{K\alpha_t \mathbf{z}_t^\ell \odot \mathbf{x}^\ell + (\alpha_{t|s} - \alpha_t)\mathbf{z}_t^\ell}{K\alpha_t \langle \mathbf{z}_t^\ell, \mathbf{x}^\ell \rangle + 1 - \alpha_t} \right.$$
$$\left. + \frac{(\alpha_s - \alpha_t)\mathbf{x}^\ell + (1 - \alpha_{t|s})(1 - \alpha_s)\mathbf{1}/K}{K\alpha_t \langle \mathbf{z}_t^\ell, \mathbf{x}^\ell \rangle + 1 - \alpha_t} \right), \quad (10)$$

and the approximate reverse posterior factorizes as per (6).

**Training** Sahoo et al. (2025a) derives a lower-variance NELBO which decomposes into a sum of token-level losses:

$$\mathcal{L}_{\text{Duo}}^{\text{NELBO}}(q, p_\theta; \mathbf{x})$$

$$= \mathbb{E}_{t \sim \mathcal{U}[0,1], q_t} \sum_{\ell=1}^{L} -\frac{\alpha_t'}{K\alpha_t} \left[ \frac{K}{\bar{\mathbf{x}}^\ell{}_i} - \frac{K}{(\bar{\mathbf{x}}_\theta^\ell)_i} \right.$$

$$- \left( \kappa_t \mathbb{1}_{\mathbf{z}_t^\ell = \mathbf{x}^\ell} + \mathbb{1}_{\mathbf{z}_t^\ell \neq \mathbf{x}^\ell} \right) \sum_{j=1}^{L} \log \frac{(\bar{\mathbf{x}}_\theta^\ell)_i}{(\bar{\mathbf{x}}_\theta^\ell)_j}$$

$$- K \frac{\alpha_t}{1 - \alpha_t} \log \frac{(\bar{\mathbf{x}}_\theta^\ell)_i}{(\bar{\mathbf{x}}_\theta^\ell)_m} \mathbb{1}_{\mathbf{z}_t^\ell \neq \mathbf{x}^\ell}$$

$$\left. - \left( (K-1)\kappa_t \mathbb{1}_{\mathbf{z}_t^\ell = \mathbf{x}^\ell} - \frac{1}{\kappa_t} \mathbb{1}_{\mathbf{z}_t^\ell \neq \mathbf{x}^\ell} \right) \log \kappa_t \right], \quad (11)$$

where $m$ and $i$ denote the indices in $\mathbf{x}$ and $\mathbf{z}_t$ respectively s.t. $\mathbf{x}_m = 1$ and $(\mathbf{z}_t)_i = 1$, $\kappa_t = (1 - \alpha_t)/(K\alpha_t + 1 - \alpha_t)$, $\bar{\mathbf{x}}^\ell = K\alpha_t \mathbf{x} + (1 - \alpha_t)\mathbf{1}$, $\bar{\mathbf{x}}_\theta^\ell = K\alpha_t \mathbf{x}_\theta(\mathbf{z}_t, t) + (1 - \alpha_t)\mathbf{1}$ and $\mathbf{x}_\theta : \mathcal{V}^L \times [0,1] \to (\Delta^K)^L$ is a shorthand for the denoising model $\mathbf{x}_\theta(\mathbf{z}_t, t)$ that uses bidirectional attention. Unlike MDMs, conditioning the USDM backbone on the diffusion time $t$ improves the validation perplexity and sample quality (Lou et al., 2024; Sahoo et al., 2025a).

### 2.3. Esoteric Language Models

> **Pro:** Fast sampling via KV caching.
> **Con:** Worse perplexity than MDLMs in diffusion mode.

Esoteric Language Model (Eso-LM) (Sahoo et al., 2025b) is a hybrid model of AR and MDLM. It closes the perplexity between AR and MDLM by smoothly interpolating between their perplexities. Unlike block diffusion (Arriola et al., 2025), it supports KV caching without sacrificing parallel generation. The marginal likelihood of this hybrid generative process is:

$$p_\theta(\mathbf{x}) = \sum_{\mathbf{z}_0 \in \mathcal{V}^L} p_\theta^{\text{AR}}(\mathbf{x} \mid \mathbf{z}_0) p_\theta^{\text{MDLM}}(\mathbf{z}_0), \quad (12)$$

where $p_\theta^{\text{MDLM}}$ is the MDLM component that generates a partially masked sequence $\mathbf{z}_0 \in \mathcal{V}^L$ in parallel, and $p_\theta^{\text{AR}}$ is the AR component that unmasks the remaining mask tokens sequentially in a left-to-right manner. The exact likelihood $\log p_\theta(\mathbf{x})$ is intractable, but Sahoo et al. (2025b) derives a variational bound:

$$-\log p_\theta(\mathbf{x}) \leq \mathbb{E}_{\mathbf{z}_0 \sim q_0} \left[ \underbrace{-\sum_{\ell \in \mathcal{M}(\mathbf{z}_0)} \log \langle \mathbf{x}_\theta^\ell(\mathbf{x}^{<\ell} \| \mathbf{z}_0^{\geq \ell}), \mathbf{x}^\ell \rangle}_{\text{AR loss}} \right]$$
$$+ \underbrace{\mathbb{E}_{q_t, t \in [0,1]} \left[ \frac{\alpha_t'}{1 - \alpha_t} \sum_{\ell \in \mathcal{M}(\mathbf{z}_t)} \log \langle \mathbf{x}_\theta^\ell(\mathbf{z}_t), \mathbf{x}^\ell \rangle \right]}_{\text{MDLM loss}}, \quad (13)$$

where $\mathbf{x}_\theta : \mathcal{V}^L \to (\Delta^K)^L$ is the shared denoising model used by both $p_\theta^{\text{AR}}$ and $p_\theta^{\text{MDLM}}$, and $q_0$ is the posterior distribution over masked sequences $\mathbf{z}_0$. This posterior is approximated by MDLM and is set to $q_0(\mathbf{z}_0|\mathbf{x})$ as defined in (4). The hyperparameter $\alpha_0 \in [0, 1]$ denotes the expected fraction of tokens in $\mathbf{x}$ generated by $p_\theta^{\text{MDLM}}$. The AR term in (13) computes the autoregressive loss over the masked positions $\mathcal{M}(\mathbf{z}_0)$ in $\mathbf{z}_0$. We write $\mathbf{x}^{<\ell} \| \mathbf{z}_0^{\geq \ell}$ for the concatenation of the sequences $\mathbf{x}^{<\ell}$ and $\mathbf{z}_0^{\geq \ell}$. This construction ensures that, when computing the autoregressive loss for the mask token at position $\ell$, all mask tokens in its left context are replaced with clean tokens. Equivalently, $\mathbf{x}^{<\ell} \| \mathbf{z}_0^{\geq \ell}$ corresponds to $\mathbf{z}_0$ with all mask tokens to the left of position $\ell$ replaced by their clean counterparts.

Eso-LM interpolates between AR and MDLM by modulating $\alpha_0$: when $\alpha_0 = 1$, $\mathbf{z}_0$ has no mask tokens and $\mathcal{L}_{\text{NELBO}}$

reduces to the MDLM's NELBO in (7); when $\alpha_0 = 0$, $\mathbf{z}_0$ is fully masked and $\mathcal{L}_{\text{NELBO}}$ reduces to the AR loss in (1). Sahoo et al. (2025b) show empirically that $\alpha_0 = 1$ is the best choice during training, since the resulting model generalizes well to a wide range of $\alpha_0$ values at inference time. Therefore, **we only consider Eso-LM in its full diffusion mode ($\alpha_0 = 1$) in this paper**.

**Forward and Reverse Processes**   Eso-LM uses the same forward and reverse processes as that of MDLM as described in Sec. 2.2.1.

**Training**   Eso-LM is trained using the low-variance objective in (8) and evaluated using the exact NELBO in (7). Rather than a bidirectional denoiser, it exploits the connection between MDMs and AO-ARMs (Ou et al., 2025) to use a decoder-only denoiser with causal attention applied to a shuffled $\mathbf{z}_t$. During training, $\mathbf{z}_t$ is shuffled so that clean tokens appear before mask tokens, with both subsets randomly permuted; positional embeddings and the corresponding ground-truth tokens are permuted in the same way. At inference time, the model fixes a generation order, which for the ancestral sampler is a random permutation of token positions. This order determines which positions are unmasked at each step, and the model iteratively unmasks tokens until the full sequence is generated. Because the denoiser uses causal attention, previously denoised tokens are independent of future tokens. This property enables KV caching while retaining parallel generation, leading to fast sampling. The downside is that replacing bidirectional attention with sparser causal attention degrades modeling capacity. As a result, in full diffusion mode, Eso-LM attains worse perplexity than MDLM.

**Block Sampler**   Sahoo et al. (2025b) found that BD3-LMs (Arriola et al., 2025) produce degenerate samples at low sampling steps due to decoding consecutive tokens in parallel. Inspired by the discovery, Sahoo et al. (2025b) proposes a sampler for Eso-LM that improves upon the ancestral sampler in MDLM, called the *Block sampler*, that only decodes far-apart tokens in parallel and significantly improves sample quality at low sampling steps. Essentially, at every denoising step $i$, the model predicts all tokens that are $L'$ apart, i.e., tokens at $\{i, i+L', \ldots, i+(k-1)L'\}$ where $k = L'/L$ and $L'$ is assumed to perfectly divide the context length $L$. Here, $k$ denotes the number of tokens denoised at each denoising step.

## 3. Scaling Laws

In this section, we establish scaling laws for state-of-the-art discrete diffusion models, including the Masked diffusion language model (MDLM; Sahoo et al. (2024a)), the uniform-state diffusion model (Duo; Sahoo et al. (2025a)),

and the interpolating diffusion model (Eso-LM; Sahoo et al. (2025b)). All comparisons are conducted under matched training FLOPs, enabling direct and fair evaluation across model families.

**Model Architecture**   All models use the Diffusion Transformer (DiT) backbone (Peebles & Xie, 2023), with rotary positional embeddings (Su et al., 2023) and adaptive layer normalization for conditioning on diffusion time (with learnable parameters). Our AR baseline replaces bidirectional attention with causal self-attention and is trained with the standard next-token negative log-likelihood (1). To match parameter counts with diffusion models, we retain the time-conditioning mechanism but set the diffusion-time input to zero for AR. For MDMs, we use bidirectional attention and likewise set the diffusion-time input to zero. USDMs also use bidirectional attention, but their reverse dynamics depend more explicitly on the noise level because tokens can transition among all vocabulary states rather than through an absorbing mask, consistent with prior work (Sahoo et al., 2025a; Lou et al., 2024; Schiff et al., 2025). As described in Sec. 2.3, we train Eso-LM in the full-diffusion setting using (8) with the denoising transformer using causal attention on the randomly shuffled input.

**Data, Tokenizer, and Context Length**   Scaling-law estimation can be distorted by data scarcity, so we follow the large-data regime advocated by compute-optimal training (Hoffmann et al., 2022). All models are trained on SlimPajama (Soboleva et al., 2023), which is sufficiently large for the compute ranges we study. We use the same tokenizer across models, specifically the Llama-2 tokenizer (Touvron et al., 2023), and a fixed context length of 2048 tokens to eliminate confounding effects from preprocessing or sequence length. We extend the vocabulary with a special mask token for all models, resulting in a total vocabulary size of 32,001. The batch size is 256.

**Compute budget**   We compute exact training compute (combined forward and backward FLOPs) using the `calflops` Python package (xiaoju ye, 2023). This contrasts with prior work (Nie et al., 2025a; Kaplan et al., 2020; Hoffmann et al., 2022), which commonly approximates training compute using $C \approx 6ND$, where $N$ is the number of non-embedding parameters and $D$ is the number of training tokens.

**Optimizer**   We use AdamW (Loshchilov & Hutter, 2019) with $\beta_1 = 0.9$, $\beta_2 = 0.95$, and weight decay 0.1. We apply a cosine learning-rate schedule with peak learning rate $4 \times 10^{-4}$ and minimum learning rate $2 \times 10^{-5}$.

*Table 1.* **Likelihood Evaluation.** SMDM-1B was trained on `slim-pajama`, while LLaDa-8B-Base was trained on proprietary data comprising 2.3T tokens. [†] Models were trained for approximately 2.1T tokens on Nemotron-Pre-Training-Dataset. Because prior work uses models of different sizes, training data, and token budgets, the results are not directly comparable to ours and are included only for reference. Underline denotes the best accuracy (↑) across all models and the **bolded** numbers denote the best diffusion accuracy.

|  | ARC-e | BoolQ | OBQA | PIQA | RACE | SIQA |
|---|---|---|---|---|---|---|
| *Chance* | *24.7* | *50.4* | *26.6* | *51.6* | *24.2* | *32.2* |
| *Prior Work* | | | | | | |
| SMDM-1B[*] (Nie et al., 2025a) | 37.4 | 61.5 | 27.0 | 60.3 | 29.3 | 37.9 |
| LLaDa-8B-Base (Nie et al., 2025b) | - | - | - | 74.4 | - | - |
| *Ours* | | | | | | |
| AR-1.7B[†] *(Autoregressive)* | 72.7 | 71.9 | 40.4 | 78.1 | 36.2 | 41.9 |
| MDLM-1.7B[†] *(Masked Diffusion)* | 50.5 | **62.8** | 32.0 | 62.2 | 34.7 | **39.2** |
| Eso-LM-1.7B[†] *(Interpolating Diffusion)* | 46.0 | 53.4 | 29.6 | 55.6 | 26.1 | 36.1 |
| Duo-1.7B[†] *(Uniform-state Diffusion)* | **53.4** | 59.6 | **33.0** | **62.7** | **35.0** | 39.0 |

## 3.1. IsoFLOP Analysis

We perform an IsoFLOP study (Hoffmann et al., 2022) over compute budgets $C \in \{6 \times 10^{18}, 1 \times 10^{19}, 3 \times 10^{19}, 6 \times 10^{19}, 1 \times 10^{20}\}$. For each target budget $C$, we train a grid of models spanning parameter counts $N$ (as in Table 4) and training-token counts $D$, producing validation losses $\mathcal{L}(N, D)$ at approximately fixed compute.

At fixed $C$, we fit a second-order model in $\log N$ to estimate the compute-optimal parameter scale:

$$\log \mathcal{L}(N; C) \approx a_C (\log N)^2 + b_C \log N + c_C, \quad (14)$$

as shown in dotted lines in Fig. 2 and define

$$N_C^* \triangleq \arg \min_N \mathcal{L}(N; C), \qquad \mathcal{L}_C^* \triangleq \mathcal{L}(N_C^*; C). \quad (15)$$

Here, $N_C^*$ is the compute-optimal parameter count at budget $C$, and $\mathcal{L}_C^*$ is the corresponding optimal validation loss. We apply this procedure identically to AR, MDLM, USDM, and Eso-LM training. As shown in Fig. 2, the diffusion models exhibit well-behaved IsoFLOP curves, comparable to those of AR models.

## 3.2. Fitting Scaling Laws

We study how (i) the compute-optimal validation loss ($\mathcal{L}_C^*$) and (ii) the compute-optimal model size ($N_C^*$) vary with training compute. From the IsoFLOP sweep we extract pairs $(C_i, \mathcal{L}_{C_i}^*)$ and fit a log-linear power law

$$(\alpha^*, \beta^*) = \arg \min_{\alpha, \beta} \sum_{i=1}^{n} \left( \log \mathcal{L}_{C_i}^* - \alpha \log C_i - \beta \right)^2, \quad (16)$$

which corresponds to $\mathcal{L}_C^* \approx \exp(\beta^*) C^{\alpha^*}$. Fig. 3a plots $\mathcal{L}_C^*$ versus $C$. Similarly, we fit a power law for the compute-optimal model size, $\log N_C^* \approx \gamma \log C + \delta$, and plot $N_C^*$ versus $C$ in Fig. 3b.

### 3.2.1. AUTOREGRESSIVE MODELS

The AR baseline recovers the familiar compute-optimal behavior: both $\mathcal{L}^*(C)$ and $N^*(C)$ follow approximate power laws over the explored budgets (Kaplan et al., 2020; Hoffmann et al., 2022). We use AR as the reference curve when comparing exponents and constant-factor gaps.

### 3.2.2. MASKED DIFFUSION MODELS

Applying the same IsoFLOP protocol to MDMs shows that their validation loss decreases approximately as a power law in compute, with a slope in log-log space comparable to AR. When trained with the true ELBO (7), we reproduce the findings of Nie et al. (2025a): MDLM requires $\approx 16\times$ more compute to match AR validation loss (Fig. 6a) and the best MDM checkpoints typically require $\approx 2\times$ fewer parameters than the AR counterparts (Fig. 6b).

**Low Variance Training Loss** We find that training MDMs with the low-variance loss (8) (while evaluating with the correct likelihood (7)) improves scaling: MDLM then requires $\approx 14\times$ (instead of $\approx 16\times$) more compute than AR to match validation loss, an approximately $12\%$ improvement in compute efficiency. A further benefit is that the compute-optimal model size decreases relative to MDLM trained with the true NELBO. We compare compute-optimal model sizes for the low-variance objective versus the true ELBO in Fig. 6c. This is notable because smaller models reduce sampling cost at inference time.

### 3.2.3. UNIFORM-STATE DIFFUSION MODELS

Duo shares the same bidirectional denoising backbone as MDLM. Under identical compute-budgeted IsoFLOP sweeps and scaling fits, we find that Duo requires $\approx 23\times$ more compute to match the AR model's perplexity. Despite weaker likelihood scaling, Duo can offer faster inference

via few-step generation enabled by self-correction; we analyze this in Sec. 3.3. The compute-optimal Duo models are also $\approx 2\times$ smaller than their AR counterparts at matched compute budget $C$.

### 3.2.4. ESOTERIC LANGUAGE MODELS

Eso-LM interpolates between MDLM and AR behavior via $\alpha_0 \in [0, 1]$, where $\alpha_0 = 0$ is fully autoregressive and $\alpha_0 = 1$ is fully diffusion. As described in Sec. 2.3, we train Eso-LM in the full-diffusion mode ($\alpha_0 = 1$). Fig. 3 shows that full-diffusion Eso-LM requires $\approx 32\times$ more compute than AR to match perplexity. Although this gap is large, Eso-LM offers a practical advantage over AR, MDLM, and Duo by supporting KV caching. The compute-optimal Eso-LM models are $\approx 2\times$ smaller than AR at matched compute budget $C$.

### 3.3. Speed-Quality Tradeoff

Scaling laws based on likelihood do not capture practical advantages at sampling time. For example, Duo supports few-step generation, and Eso-LM support KV caching capabilities not reflected in validation perplexity alone. We therefore study the speed–quality tradeoff across methods.

For each compute budget $C \in \{6 \times 10^{18}, 1 \times 10^{19}, 3 \times 10^{19}, 6 \times 10^{19}, 1 \times 10^{20}\}$, we select the compute-optimal model. We sample autoregressively from the AR model; for MDLM and Duo we use the ancestral sampler and vary the number of sampling steps via $T$; and for Eso-LM we use the Block sampler described in Sec. 2 and vary $L'$. For evaluation, we draw 1000 unconditional samples and compute Generative Perplexity (Gen PPL) under a pretrained Llama-2 (7B) model. Lower Gen PPL indicates higher-quality samples. We also report throughput (tokens/sec; higher is better). To measure throughput, we use the largest power-of-two batch size that fits on a single 80GB H100 GPU for each model size.

**Relating Throughput and Quality**  Sample quality in diffusion LLMs can be improved by either increasing the denoiser size or increasing the number of sampling steps $T$. To compare these tradeoffs, we fit Gen PPL and throughput as functions of $T$:

$$\text{Gen PPL}(T; \alpha_C, \beta_C, \gamma_C) = \alpha_C + \beta_C T^{\gamma_C} \quad (17)$$

$$\text{Throughput}(T; \alpha'_C, \beta'_C, \gamma'_C) = \alpha'_C + \beta'_C T^{\gamma'_C} \quad (18)$$

where $\alpha_C, \alpha'_C, \beta_C, \beta'_C, \gamma_C, \gamma'_C \in \mathbb{R}$. Fig. 4 shows throughput curves and fits, while Fig. 5b reports Gen PPL and average sequence entropy as a measure of diversity (Zheng et al., 2024; Sahoo et al., 2025a). Following Zheng et al. (2024), we perform all sampling in float64 precision to avoid artificially low diversity and Gen PPL. Under high-precision sampling, sequence entropy remains stable across diffusion steps.

**Speed-Quality Pareto Frontier**  To construct a speed-quality Pareto frontier, we proceed as follows. For each method and model size, we (i) use the fitted Gen PPL curve (17) to compute the number of steps $T$ required to reach a target Gen PPL, (ii) evaluate the corresponding throughput using (18), and (iii) take the maximum throughput across model sizes. We repeat this for target Gen PPL values in $\{40, \ldots, 200\}$ and plot the resulting frontier in Fig. 1. We observe that **AR models produce the highest-quality samples but are the slowest**. For throughput $< 200$, AR dominates. **Duo dominates in throughput ranges** $[200, 400] \cup [600, \infty]$ due to few-step generation. **Eso-LM dominates in the intermediate range** $[400, 600]$ as they uniquely support KV caching among the diffusion LLMs studied here.

> **Key takeaway:** Perplexity alone can be misleading when comparing diffusion language models. Although Masked diffusion exhibits the best likelihood scaling, Uniform-state and interpolating diffusion models can be preferable in practice due to more efficient sampling.

## 4. Scaling to the Billion-Parameter Regime

We scale AR, MDLM, Duo, and Eso-LM to 1.7B parameters with a context length of 2048. All models are pretrained on 2.1T tokens using a data protocol that closely matches modern LLM training pipelines (Nie et al., 2025b; Yang et al., 2025), without introducing any specialized techniques. The corpus is drawn from large-scale online text, with low-quality content filtered via a combination of manually designed heuristics and LLM-based filtering. We train our models on the phase 1 and phase 2 mixes of the Nemotron-Pre-Training-Dataset (Basant et al., 2025) that contains general-domain text and high-quality math data.

Each model is trained on 64 H100 GPUs. To improve robustness to variable-length inputs, we follow Gulrajani & Hashimoto (2024); Nie et al. (2025b) and set 1% of pretraining sequences to a random length sampled uniformly from $\mathcal{U}[1, 2048]$. We use AdamW as in Sec. 3, with a peak learning rate of $3 \times 10^{-4}$ and a minimum learning rate of $4 \times 10^{-5}$. We linearly warm up the learning rate from 0 to $3 \times 10^{-4}$ over the first 2000 iterations, then keep it constant at $3 \times 10^{-4}$. After processing 1.4T tokens, we decay the learning rate to $4 \times 10^{-5}$ over the remaining 0.7T tokens to promote stable training. We use a global batch size of 256 and a vocabulary size of 128,000.

### 4.1. Likelihood-Based Benchmarks

We evaluate the 1.7B models in the zero-shot setting on a standard suite of likelihood-based downstream benchmarks spanning commonsense reasoning and reading com-

prehension: ARC-e (Clark et al., 2018), BoolQ (Clark et al., 2019), PIQA (Bisk et al., 2020), SIQA (Sap et al., 2019), OBQA (Mihaylov et al., 2018), and RACE (Lai et al., 2017).

**Results** As shown in Table 1, the AR model achieves the best overall performance. Among diffusion models, **MDLM performs best on ARC-e, BoolQ, and SIQA**, while **Duo leads on OBQA, PIQA, and RACE**.

### 4.2. Maths and Reasoning Benchmark

We also assess mathematical reasoning on GSM8K (Cobbe et al., 2021). Following Nie et al. (2025a), we perform supervised fine-tuning (SFT) on the augmented GSM8K dataset from Deng et al. (2024), which expands the original GSM8K training set to 385K samples using GPT-4-generated augmentations. For all models, we use AdamW with a cosine schedule (as above) and conduct a grid search over learning-rate pairs $(\eta_{max}, \eta_{min})$ (see Table 3), reporting the best setting for each method. We fine-tune each model for 5 epochs with a context length of 256.

**Results** Prior diffusion LLM work often uses confidence-based sampling (Nie et al., 2025a;b), which effectively collapses to left-to-right generation (Nie et al., 2025b). We therefore generate left-to-right one token at a time from all models. As shown in Table 2, **Duo outperforms AR, MDLM, and Eso-LM on this math-and-reasoning evaluation**. We additionally report throughput (↑), measured in tokens per second, evaluated on the full GSM8K test set (1319 examples) using a batch size of 1. Throughput is computed using generated tokens only (including EOS), excluding the prompt and any tokens following EOS. In this memory-bound setting, it is expected that AR models achieve comparable latency to diffusion models despite using KV caching; see (Liu et al., 2025).

> **Key Takeaway:** Although Uniform-state diffusion (Duo) exhibits worse likelihood scaling than Autoregressive and Masked diffusion, it surpasses them on math and reasoning after SFT.

## 5. Discussion and Conclusion

We revisit a common assumption in discrete diffusion: that Masked diffusion models (MDMs), and perplexity-based scaling analyses in particular, provide a definitive recipe for competitive diffusion language models. Through compute-matched IsoFLOP scaling across three diffusion families (Masked, Uniform-state, and interpolating), and by combining likelihood-based and sampling-centric evaluations, we show that this view is incomplete. d-LLMs can exhibit scaling exponents comparable to autoregressive models (ARMs), but with large constant-factor gaps that depend

*Table 2.* **GSM8K benchmark.** SMDM-1B is trained on `slim-pajama` while LLaDa-8B-Base on proprietary data. [†] Models are trained for 2.1T tokens on the Nemotron-Pre-Training-Dataset. Throughput (tokens/sec) is measured using a batch size of 1. In this memory-bound setting, AR models are expected to match the latency of diffusion models despite using KV caching.

| | Acc. (↑) | Throughput (toks / sec; ↑) |
|---|---|---|
| *Prior work* | | |
| SMDM-1B[*†] | 58.5 | - |
| LLaDa-8B | 70.7 | - |
| *Ours* | | |
| AR-1.7B[†] | 62.9 | 25.6 |
| MDLM-1.7B[†] | 58.8 | 24.6 |
| Eso-LM-1.7B[†] | 33.4 | 25.6 |
| Duo-1.7B[†] | **65.8** | 24.6 |

strongly on the diffusion family. MDLM shows the strongest likelihood scaling, while Duo and Eso-LM require substantially more compute to match AR perplexity. Notably, our work is the first to study the scaling laws of Uniform-state diffusion (concurrently with von Rütte et al. (2025)) and AR-MDLM interpolating diffusion. We further show that training MDLM with a low-variance objective improves compute efficiency over standard ELBO training, yielding smaller compute-optimal models and lower sampling cost.

Crucially, perplexity alone is insufficient for comparing diffusion methods across families. While it is meaningful within a family, where NELBO bounds are shared, it can be misleading across families with fundamentally different diffusion processes. **Uniform-state and interpolating models may scale worse in likelihood yet be preferable in practice due to more efficient sampling**, such as few-step generation in Duo and KV caching in Eso-LM. This motivates evaluation standards that jointly consider likelihood, sampling efficiency, and downstream performance. Scaling all methods to 1.7B parameters shows these tradeoffs persist. AR models remain strongest on likelihood-based metrics, while uniform-state diffusion (Duo) can outperform AR and other diffusion models on math and reasoning after supervised fine-tuning, despite weaker perplexity scaling.

Looking ahead, scaling diffusion LLMs to much larger sizes may enable deeper study of emergent behaviors (Wei et al., 2022a) and long-range reasoning (Wei et al., 2022b), and clarify whether their distinct generative mechanisms yield consistent real-world advantages. More broadly, this line of work may clarify how model capabilities arise and the extent to which autoregressive factorization itself underpins the strengths of modern LLMs.

## Impact Statement

This paper presents work whose goal is to advance the field of Machine Learning. There are many potential societal

consequences of our work, specifically those related to the generation of synthetic text. Our work can also be applied to the design of biological sequences, which carries both potential benefits and risks.

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

## Contents

# Appendices

## A. Additional Experiments

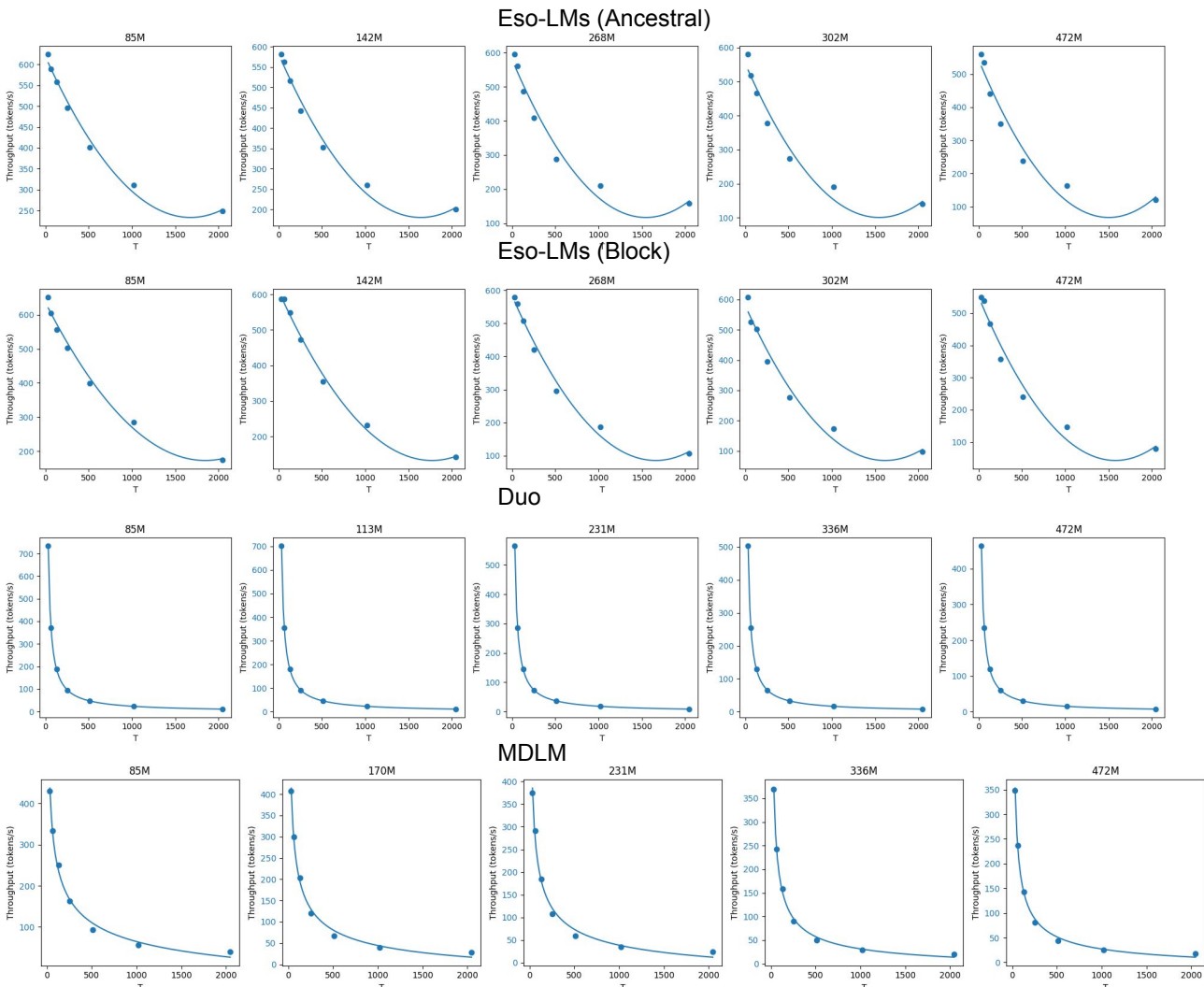

*Figure 4.* Throughput (toks / sec; ↑) vs time discretization $T$ for various diffusion models.

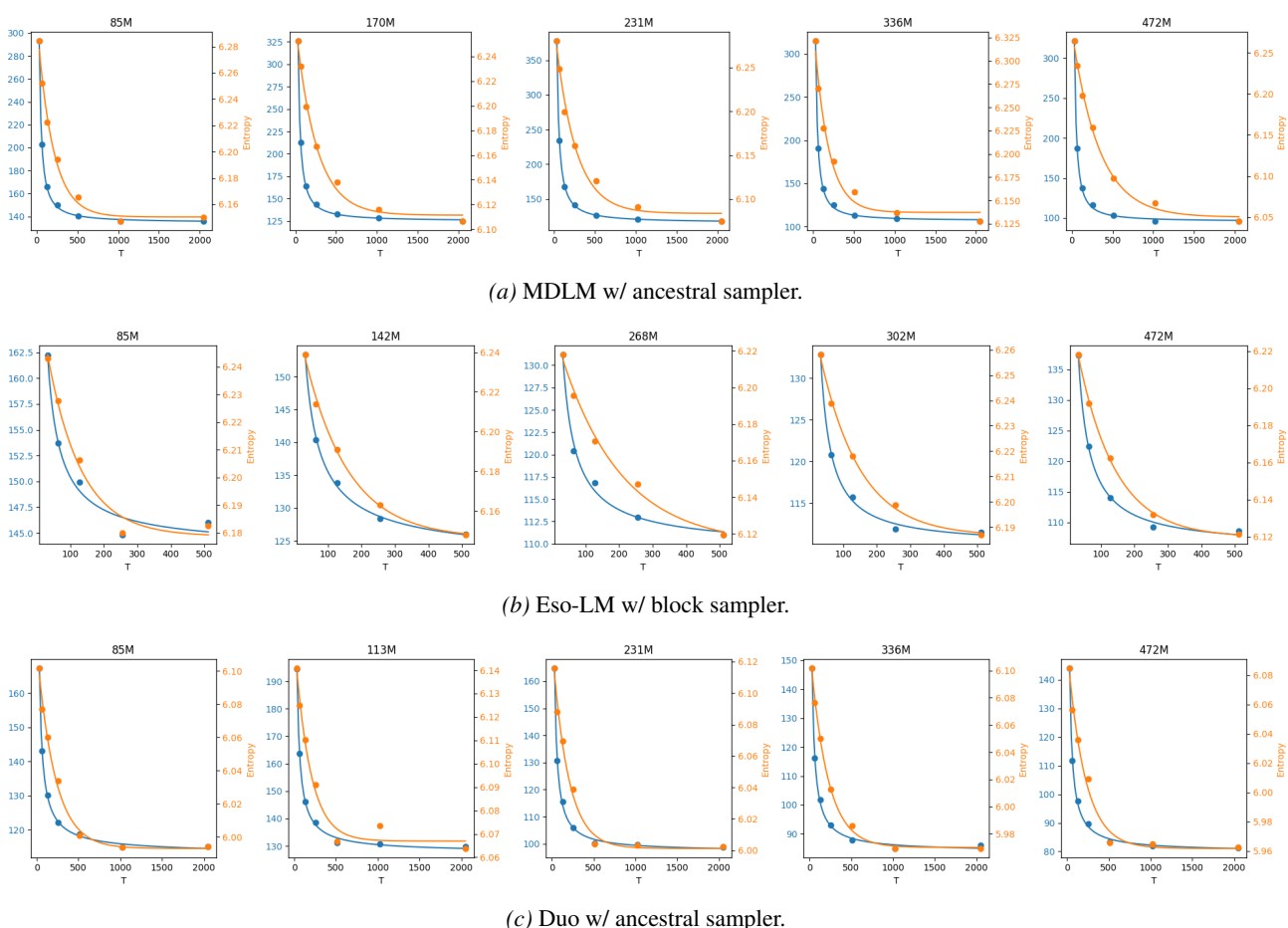

*(a)* MDLM w/ ancestral sampler.

*(b)* Eso-LM w/ block sampler.

*(c)* Duo w/ ancestral sampler.

*Figure 5.* Gen. PPL (sample quality; ↓) and entropy (sample diversity; ↑) vs time discretization ($T$) for (a) MDLM w/ ancestral sampler, (b) Eso-LM w/ block sampler, and (c) Duo w/ ancestral sampler.

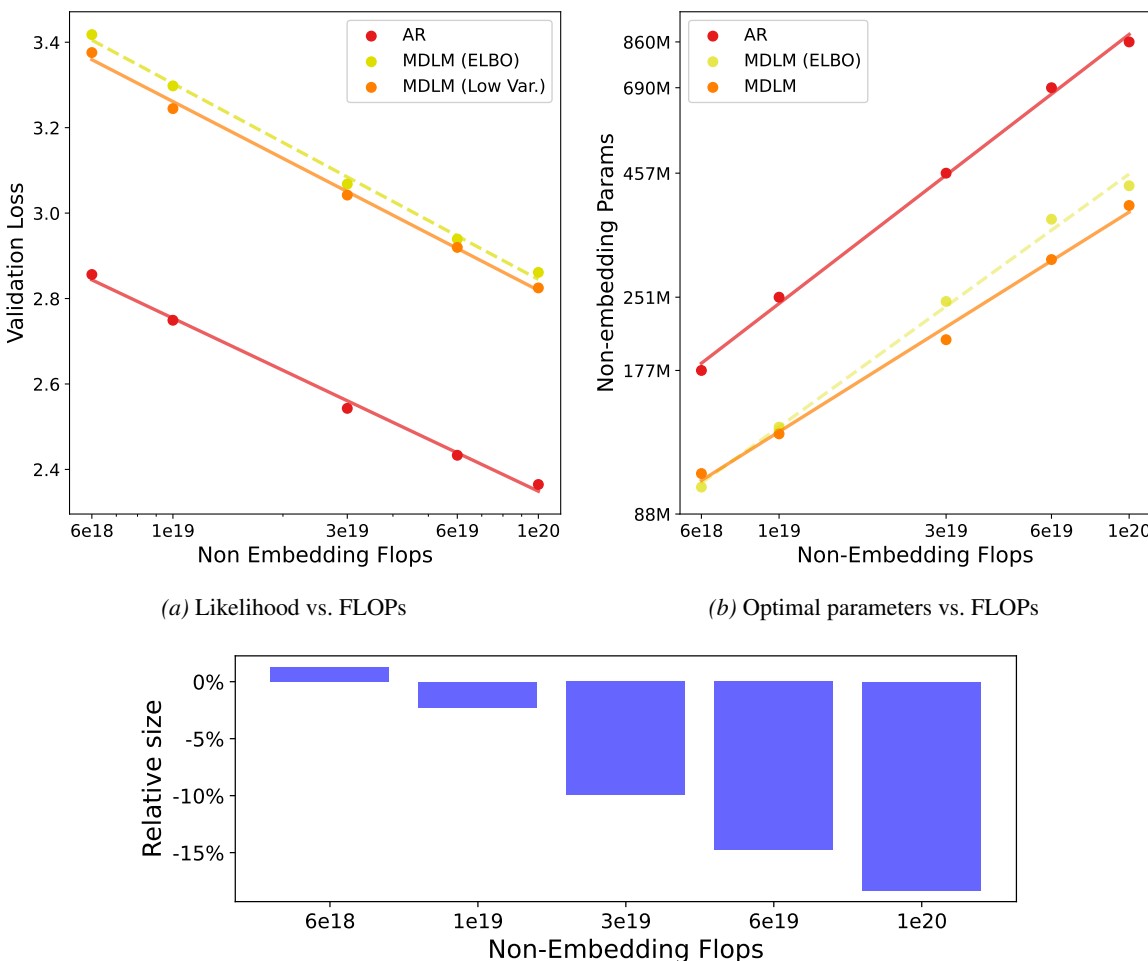

*(a)* Likelihood vs. FLOPs

*(b)* Optimal parameters vs. FLOPs

*(c)* Fractional difference in model sizes of MDLM trained with the low-variance training loss (8) relative to MDLM trained with the ELBO (7).

*Figure 6.* Scaling law comparison between MDLM trained with the low-variance training loss (8) and MDLM trained with the ELBO (7). MDLM trained with the low variance training loss yields compute optimal models with fewer parameters.

*Table 3.* Accuracy after SFT on GSM8K with a cosine schedule, with maximal and minimial learning rates $\eta_{\max}$ and $\eta_{\min}$.

| Hyperparams | | Accuracy (↑) | | |
|---|---|---|---|---|
| $\eta_{\max}$ | $\eta_{\min}$ | **5 ep** | **10 ep** | **20 ep** |
| AR | | | | |
| 1e-5 | 2e-6 | 62.9 | – | – |
| 2e-5 | 5e-8 | 59.0 | – | – |
| 2e-5 | 5e-7 | 61.6 | – | – |
| 2e-5 | 2e-6 | 59.1 | – | – |
| 4e-5 | 2e-6 | 55.0 | – | – |
| 5e-5 | 5e-6 | 50.7 | – | – |
| MDLM | | | | |
| 1e-5 | 2e-6 | 58.4 | 54.7 | 53.1 |
| 2e-5 | 5e-8 | 56.6 | 59.7 | 54.0 |
| 2e-5 | 5e-7 | 58.8 | 59.2 | 54.9 |
| 2e-5 | 2e-6 | 61.7 | – | 54.1 |
| 4e-5 | 2e-6 | 57.9 | 56.0 | 51.2 |
| 5e-5 | 5e-6 | 55.3 | 53.4 | 49.6 |
| *Eso-LM* | | | | |
| 2e-5 | 2e-6 | 33.4 | – | – |
| Duo | | | | |
| 1e-5 | 2e-6 | 64.6 | 64.8 | 61.8 |
| 2e-5 | 5e-8 | 66.0 | 65.4 | 60.2 |
| 2e-5 | 5e-7 | 65.8 | 64.4 | 59.8 |
| 2e-5 | 2e-6 | 65.0 | 63.0 | 58.4 |
| 4e-5 | 2e-6 | 62.9 | 59.0 | 53.2 |
| 5e-5 | 5e-6 | 60.5 | 54.8 | 51.2 |

*Table 4.* Transformer configurations of AR, MDLM, Duo, and Eso-LM used in the scaling law study.

| Non-Embedding Parameters ($M$) | n_embed | n_layers | n_heads |
|:---:|:---:|:---:|:---:|
| 14 | 256 | 6 | 4 |
| 29 | 384 | 8 | 6 |
| 44 | 512 | 8 | 8 |
| 58 | 576 | 9 | 9 |
| 74 | 640 | 10 | 10 |
| 91 | 640 | 13 | 10 |
| 107 | 640 | 16 | 8 |
| 116 | 768 | 12 | 12 |
| 140 | 768 | 15 | 12 |
| 163 | 768 | 18 | 12 |
| 173 | 896 | 14 | 14 |
| 194 | 896 | 16 | 14 |
| 214 | 896 | 18 | 14 |
| 247 | 1024 | 16 | 16 |
| 274 | 1024 | 18 | 16 |
| 300 | 1024 | 20 | 16 |
| 413 | 1280 | 18 | 10 |
| 475 | 1280 | 21 | 10 |
| 493 | 1408 | 18 | 11 |
| 537 | 1280 | 24 | 10 |
| 568 | 1408 | 21 | 11 |
| 642 | 1408 | 24 | 11 |
| 698 | 1536 | 22 | 12 |
| 787 | 1536 | 25 | 12 |
| 1016 | 1792 | 24 | 14 |
| 1208 | 2048 | 22 | 16 |
| 1364 | 2048 | 25 | 16 |
| 1708 | 2176 | 28 | 17 |

