# OpenReview forum: "Scaling Beyond Masked Diffusion Language Models"
_ICML.cc/2026/Conference — ICML 2026 regular_

### Official Review · Reviewer_FghQ · 2026-02-21

**Soundness:** 2
**Presentation:** 2
**Significance:** 2
**Originality:** 2
**Overall Recommendation:** 4
**Confidence:** 3

**Summary:**

This paper presents the systematic IsoFLOP scaling study comparing three families of discrete diffusion language models (masked, uniform-state, and interpolating) under matched compute budgets, demonstrating that perplexity alone is insufficient for cross-family evaluation and that non-masked diffusion models can offer practical advantages in specific scenarios when assessed holistically via sampling efficiency and downstream task performance.

**Compliance With Llm Reviewing Policy:**

Affirmed.

**Final Justification:**

The authors have addressed my concerns, especially regarding data reproducibility; therefore, I have increased my score.

**Key Questions For Authors:**

Please see Weaknesses above

 I will also take into account the opinions of other reviewers and make a comprehensive decision

**Limitations:**

yes

**Strengths And Weaknesses:**

Strengths：

1. Challenges perplexity-centric evaluation: Shows models with worse likelihood scaling can outperform on downstream tasks and sampling efficiency.

2. Practical contribution: Simple cross-entropy training objective improves MDM FLOPs efficiency by ~12% while preserving correct likelihood evaluation.

Weaknesses：

1. Scale limitation: Max 1.7B params;  unclear if findings generalize to 8B+ LLMs.

2. Reproducibility gap: Large-scale (1.7B) results use proprietary data, limiting full replication.

3. Lack of novelty: The current experimental results are not very inspiring to think, and feel more engineering-oriented.

---

> ### Author Rebuttal · Authors · 2026-03-31
>
> We thank the reviewer for the thoughtful feedback and for acknowledging that the paper
> 1. **Challenges the metrics currently used to compare categorical diffusion LLMs in the literature**, as well as
> 2. **Its contribution to improving the training efficiency of MDLM**.
>
> We address the concerns below.
> # Concern 1: Scale limitation
>
> > Max 1.7B params; unclear if findings generalize to 8B+ LLMs.
>
> We agree that larger-scale experiments at 8B+ would be valuable, but they are infeasible within our resource constraints. Our study spans four paradigms (AR, MDLM, Duo, Eso-LMs), and for MDLM we evaluate two loss formulations. Scaling experiments alone required ~50,000 H100 hours. **Training four 1.7B models required 250,000 H100 hours ($1M USD on Lambda cloud computing service)**.
>
> **We estimate that Scaling each of these models at 7B scale would require** 600000 H100 Hours (**> 2 Million USD**) which is beyond the capacity of our lab.
>
> ---
>
> # Concern 2: Lack of Novelty
> > The current experimental results are not very inspiring to think, and feel more engineering-oriented.
>
> We respectfully disagree that the paper lacks novelty or is merely engineering-oriented. Its central contribution is a conceptual one: **it reveals that the metrics currently used to evaluate and compare diffusion LLMs can be fundamentally misleading**. As a result, the paper does more than report an incremental empirical result; **it challenges the assumptions that are currently shaping the direction of the field**.
>
> This matters because the community has **increasingly concentrated on scaling masked diffusion models** [1, 2], largely due to the belief that other diffusion formulations, such as uniform-state diffusion, are inferior. **Our findings overturn that narrative.** Despite having worse perplexity, Uniform-state diffusion (Duo) outperforms MDLM on math and reasoning tasks. This shows that perplexity alone is not a sufficient proxy for downstream capability, and suggests that **the field may be prioritizing the wrong model families based on incomplete evaluation criteria**.
>
> **We therefore believe the paper is both novel and impactful**: it compels the community to reconsider not only how diffusion language models should be evaluated, but also which directions are most promising for their future development.
>
>
> [1] Nie et al., “Large Language Diffusion Models”, 2025.
>
> [2] Bie et al., “LLaDA2.0: Scaling Up Diffusion Language Models to 100B”, 2025.
>
> ---
>
> # Comment
> > on training the model on proprietary data
>
> We thank the reviewer for pointing this out. We would like to clarify that the statement indicating the use of proprietary data was a typographical error. All models in our experiments were trained exclusively on the publicly available Nemotron-Pre-Training-Dataset [1] (on the phase 1 and phase 2 mixes). We will correct this wording in the final version.
>
> [1] Basant et al., “Nvidia nemotron nano 2: An accurate and efficient hybrid mamba-transformer reasoning model.” 2025

---

> > ### Author Rebuttal · Reviewer_FghQ · 2026-04-01
> >
> > Thank you for the authors’ response. My concerns have been addressed, and I will increase my score from 3 to 4.

---

> > > ### Author Response · Authors · 2026-04-06
> > >
> > > We thank the reviewer for finding our clarifications valuable and for increasing their score. We are also grateful for their careful review and constructive feedback on the paper.

---

### Official Review · Reviewer_kFfu · 2026-03-11

**Soundness:** 3
**Presentation:** 2
**Significance:** 2
**Originality:** 2
**Overall Recommendation:** 2
**Confidence:** 4

**Summary:**

This paper studies and compares the scaling laws and performance of 4 different models: ARs, MDMs, USDMs, and Eso-LMs. It evaluates the validation loss over different values of compute budget and number of parameters. In addition, it proposed to use other metrics for evaluating diffusion models such as token throughput, speed-quality ratio, and extends these models to 1.7B to evaluate them how they perform on downstream tasks. The paper, like prior work, concludes that USDMs is a promising and often ignored alternative, and that the best model in terms of generative perplexity depends on the desired throughput,

**Compliance With Llm Reviewing Policy:**

Affirmed.

**Final Justification:**

I go through the unaddressed points below.

*We could not verify the claim that was publicly visible in September 2025.*

Since the work was accepted at ICLR 2026, and the full submission deadline was September 24, the submission became public immediately thereafter. While the camera-ready version can of course include some changes, the central message of a paper typically does not change substantially at that stage. In addition, the original submission is ordinarily accessible through the 'revisions' tab, although for 2026 there appears to be a technical issue at present. In any case, the paper was publicly available roughly four months before the ICML 2026 deadline.

There appears to have been some misunderstanding in the rebuttal regarding the scope of my concern. At no point did I argue that the paper lacks novelty merely because von Rütte et al. exists [1]. My request was more limited and, I believe, standard: the paper should more clearly acknowledge this prior work as prior work, and compare against it in order to clarify what is new in the present submission, and what overlaps. In my view, the current submission does contain several new results. My concern has been simply that these contributions should be positioned more carefully within the existing literature and that attribution should be made explicit. The submission now contains a useful starting point for doing so:

'''At 1.7B parameters, Duo beats AR and MDLM on maths/reasoning, and our speed–quality analysis shows that models with worse likelihood scaling can still be preferable in practice.
We study the scaling laws for AR–MDLM interpolation regime with Eso-LMs, which Rutte et al lacks.
We making MDLM training 12% FLOP-efficient unlike Rutte et al.'''

At present, I do not feel comfortable recommending acceptance as the authors oppose acknowledging prior work.

*The review applies an inconsistent standard: it accepts an essentially ELBO-driven argument in von Rütte et al. as sufficient to motivate uniform diffusion, but labels our paper “unqualified” even though our paper explicitly argues that likelihood alone is insufficient and adds speed–quality and downstream evidence precisely to address that limitation.*

My evaluation is of the present submission, not von Rütte et al. [1]. The point is simply that some ideas and conclusions appear to overlap across the two papers, and that this overlap should be acknowledged through citation and discussion. To the extent that the two works may share certain weaknesses, I can only comment on those insofar as they affect the submission currently under review.

*This comment selectively quotes the paper and, in doing so, mischaracterizes our claim.*

For reference, here is the full claim with context:

"Under identical compute-budgeted IsoFLOP sweeps and scaling fits, we find that Duo requires ≈ 23× more compute to match the AR model’s perplexity. Despite weaker likelihood scaling, Duo can offer faster inference via few-step generation enabled by self-correction; we analyze this in Sec. 3.3."

My concern remains about how these statements are phrased. The paper states that Duo requires more compute "to match the AR model's perplexity", but for Duo we only have a bound, not the true perplexity itself. By the point where the two quantities become equal, we therefore do not know whether Duo's true perplexity is actually much lower than AR's or not. Similarly, the phrase "Despite weaker likelihood scaling," seems to rely on a quantity that is only accessible through a bound in Duo's case. I think these statements would be much clearer if the paper explicitly noted, at the point where they are made, that the relevant quantities are bounds being used as proxies for the true likelihood. As written, I find the wording potentially misleading. This is not a criticism unique to this paper, since the limitation is a broader unresolved issue in the field, but it does affect how the paper's claims should be presented.

*MAUVE Scores*

I found the new MAUVE results informative and useful. In particular, they suggest a meaningful difference in mode distribution between AR and non-AR methods, especially in how performance changes as throughput increases. In my view, these results are important enough to be included in the paper, and they would also have strengthened the rebuttal.

Overall, the main requests from my original review that, in my view, remain insufficiently addressed are the following:

1. The current submission should be positioned more clearly relative to von Rütte et al. [1].
2. The MAUVE scores should be included in the paper.
3. The perplexity claims should be revised so that they are correct in context.

[1] von Rütte et al., 2025.

**Key Questions For Authors:**

**Q1.** Could the authors explain the difference between USDMs and Duo?

**Q2**. Can the authors calculate MAUVE as well for experiments in Figures 5, 6, 7?

**Limitations:**

There is no particular limitation section, but the paper does state that scaling diffusion LLMs to much larger sizes may enable deeper study of emergent behaviors and long-range reasoning and clarify whether their distinct generative mechanisms yield consistent real-world advantages.

**Strengths And Weaknesses:**

**Strengths**

**S1.** The paper is clearly written and structured.

**S2.** The paper addresses an interesting problem.

**S3.** The paper claims appear to be sound and well backed by empirical evidence. In particular I appreciate Figure 2, and I believe it can be useful for the community.

**Weaknesses**

**W1.**  One of the main weaknesses of the paper is the insufficient distinguishing of the proclaimed contributions from prior work:

a) [1] already makes the claim that Uniform diffusion is underappreciated and has nice scaling properties. In fact, despite that paper studying precisely the scaling laws of diffusion LLMs, there is no mention of it in the paper. The paper needs to be modifies significantly to include this is prior work and explain the differences between [1] and current work.

b) In line 185 right column, this submission credits [3] for the bound of USDMs. However, even [3] credits [4] for it (line above Equation 4, [4].

c) Line 253-254 credits again Sahoo et al. (2025a) for using the simplified cross entropy, but many prior works use this loss, including $L_{ll}$ in the early [2].

**W2.** In the cases in which the generative perplexity is measured, collapse is measured only within sentences (entropy) but not mode collapse as well. Can the authors report MAUVE scores as well?

**W3.** While the term validation loss is used correctly, practically speaking, what matters is the validation perplexity or the validation NLL. The validation loss in AR and that of diffusion models are different, so it is somewhat like comparing apples to oranges. Indeed, ARs report the NLL while the other models only a bound. This makes the experiments and conclusions somewhat unfair to diffusion models. In particular the paper notes a difference in constant between AR and discrete diffusion, which could be simply the difference between the bound and the true NLL.

**Minor Weaknesses**

**M1.** The paper appears to use USDMs and Duo interchangeably. Why not simply use USDMs? In particular it does not seem like this paper makes use of duality and it does not even introduce it in its detailed background section.

**M2.** In the case of Duo, results in the appendix show a significant drop in in terms of entropy.



[1] Rutte et al. SCALING BEHAVIOR OF DISCRETE DIFFUSION LANGUAGE MODELS. 2025

[2] Campbell et al. A Continuous Time Framework for Discrete Denoising Models. 2022

[3] Sahoo et al. The diffusion duality. 2025

[4] Sciff et al.  Simple guidance mechanisms for discrete diffusion models. 2025

---

> ### Author Rebuttal · Authors · 2026-03-31
>
> We thank the reviewer for the thoughtful feedback and for
> 1. **recognizing that the paper addresses an important problem** and that
> 2. the speed-quality pareto frontier (Figure 2) is useful to the community.
>
> We address the concerns below.
>
> # Concern: Validation Perplexity
> > The validation loss in AR and that of diffusion models are different, so it is somewhat like comparing apples to oranges.
>
> This observation is correct and, in fact, serves as a key motivation for the paper. The central contribution is **revealing that the metrics currently used to evaluate and compare diffusion LLMs are fundamentally misleading**.
>
> As a result, the paper does more than report an empirical result; **it challenges the assumptions that are currently shaping the direction of the field**.
> This matters because the community has **increasingly concentrated on scaling masked diffusion models** [1, 2], largely due to the belief that other diffusion formulations, such as uniform-state diffusion, are inferior. **Our findings overturn that narrative.** Despite having worse perplexity, Uniform-state diffusion (Duo) outperforms MDLM on math and reasoning tasks. This shows that perplexity alone is not a sufficient proxy for downstream capability, and suggests that **the field may be prioritizing the wrong model families based on incomplete evaluation criteria**.
>
> [1] Nie et al., “Large Language Diffusion Models”, 2025.
>
> [2] Bie et al., “LLaDA2.0: Scaling Up Diffusion Language Models to 100B”, 2025.
>
> > In particular the paper notes a difference in constant between AR and discrete diffusion, which could be simply the difference between the bound and the true NLL.
>
>
> Indeed, this may well be the case. This is precisely why we recommend against using perplexity to compare across model families, such as masked diffusion, AR, and uniform-state models. Instead, the paper advocates for alternative evaluations, including:
>
> - **Speed–quality Pareto frontiers (Fig. 1):** Duo and Eso-LMs exhibit worse perplexity than MDLM but are Pareto superior in the speed–accuracy tradeoff.
> - **Downstream evaluations on math and reasoning tasks (Tab. 2):** Duo outperforms MDLM despite having worse perplexity.
>
> # Comment: Prior Work
>
> >  [1] already makes the claim that Uniform diffusion is underappreciated
>
> We agree that [1] makes a related high-level claim. However, [1] was made public on Dec. 25, 2025, less than one month before the ICML submission deadline, and is therefore considered concurrent work under ICML policy. More importantly, our contributions differ in substance:
> - [1] studies scaling behavior only, whereas we provide downstream evaluations showing that uniform-state diffusion can outperform masked diffusion,
> - compare a broader set of diffusion model families including the AR–MDLM interpolation regime (Eso-LMs), and
> - argue that MDLMs should be trained with a simple cross-entropy objective rather than the true NELBO.
>
>  > this submission credits [3] for the bound of USDMs.
>
> Thank you for pointing this out. We cited [3] because the NELBO form used in Eq. 9 follows [3], which gives a Rao-Blackwellized estimator with the same mean and lower variance for the NELBO in UDLM [4], which is itself a Rao-Blackwellized version of SEDD-Uniform [5]. We will clarify this lineage in the revision.
>
> > Line 253-254 credits again Sahoo et al. (2025a) for using the simplified cross entropy,
>
> The reviewer is correct that the simplified cross-entropy objective appeared earlier in [2,7]. However, those works used it as a proxy because the true NELBO for these processes had not yet been derived; that came later in [8,9]. Since then, **NELBO training has become standard for MDLM pretraining, including at 100B scale [10]**. [6] was the first to notice low variance training with the simplified cross-entropy and using it instead of optimizing the true NELBO. We will make this historical context explicit.
>
>  > The paper appears to use USDMs and Duo interchangeably. Why not simply use USDMs
>
> >  Could the authors explain the difference between USDMs and Duo?
>
> USDM refers to the broader class, while Duo is a specific uniform-state diffusion algorithm. We use “Duo” for the same reason we use “MDLM” and “Eso-LMs”: each denotes a specific algorithm.  A notable distinction is that Duo uses a different NELBO form, with lower training variance than UDLM [4] and SEDD-Uniform [5].
>
> > In the case of Duo, results in the appendix show a significant drop in in terms of entropy.
>
> In the appendix, Duo’s entropy is only 0.1 lower than MDLM and Eso-LMs (about `1.6%`), which we view as a small and acceptable trade-off.
>
> [1] Rutte et al. SCALING BEHAVIOR OF DISCRETE DIFFUSION LANGUAGE MODELS. 2025
>
> [2] Campbell et al. 2022
>
> [3] Sahoo et al. The diffusion duality. ICML 2025
>
> [4] UDLM: Schiff et al. 2025
>
> [5] SEDD: Lou et al., ICML 2024
>
> [6] Sahoo et al., “Esoteric Language Models”, 2025
>
> [7] DFM: Gat et al., ICML 2024.
>
> [8] MDLM: Sahoo et al, ICML 2025
>
> [9] MD4: Shi et al., ICML 2025

---

> > ### Author Rebuttal · Reviewer_kFfu · 2026-04-02
> >
> > Thank you for the clarifications. After reading the rebuttal, I still intend to maintain my current score.
> >
> > On novelty, I do not find the current positioning sufficiently precise. In particular, von Rutte et al.'s *Scaling Behavior of Discrete Diffusion Language Models* was already publicly visible on OpenReview as an ICLR 2026 submission in September 2025, and therefore should be treated as prior public work rather than as merely concurrent work. More importantly, that paper already studies the scaling behavior of masked and uniform discrete diffusion and explicitly argues that uniform diffusion is a promising and underexplored regime. The discussion of prior work should therefore make the distinction from that paper much more explicit. Likewise, the lineage around the likelihood / ELBO discussion should be credited more carefully. Shaul et al.'s *Flow Matching with General Discrete Paths: A Kinetic-Optimal Perspective* already provided a general ELBO / likelihood-bound perspective for discrete paths in 2024.
> >
> > On evaluation, my concern is not simply that an additional metric would be helpful. The manuscript explicitly frames diffusion training in terms of a negative variational bound / NELBO, and for Eso-LMs it further states that the exact likelihood is intractable and is replaced by a variational bound. Yet the paper also states that ``all comparisons are conducted under matched training FLOPs, enabling direct and fair evaluation across model families,'' and then makes unqualified cross-family claims such as MDLM requiring about $16\times$ more compute to match AR validation loss, Duo requiring about $23\times$ more compute to match AR perplexity, and Eso-LMs requiring about $32\times$ more compute than AR to match perplexity. I therefore do not think the issue is one of interpretation: the paper itself already compares quantities that are not fully comparable across families, and it does so without sufficient qualification.
> >
> > Relatedly, once the paper moves beyond perplexity and explicitly compares sample quality and diversity, it is unclear why it stops at Gen PPL and average sequence entropy rather than reporting MAUVE. My point is not that entropy is irrelevant; it is that, if the paper is already making diversity-oriented comparisons, MAUVE is the more appropriate metric for cross-family comparison of distributional quality and mode coverage. Stopping at entropy leaves the mode-coverage question insufficiently addressed.
> >
> > Overall, I do not think the rebuttal adequately resolves either the novelty concerns or the evaluation concerns, and I therefore intend to maintain my current score.

---

> > > ### Author Response · Authors · 2026-04-03
> > >
> > > > Rutte et al was already publicly visible on OpenReview as an ICLR 2026 submission in September 2025
> > >
> > >
> > > We could not verify the claim that was publicly visible in September 2025. The current openreview page (https://openreview.net/forum?id=GDYaNzxt9T) lists the paper as published in Jan 2026 and modified on 26 Feb 2026 with the arxiv version posted in Dec 2025. **We therefore respectfully ask the AC to assess this point** based on verifiable public records rather than an unsupported September 2025 assertion.
> > >
> > > ---
> > >
> > > >  More importantly, that paper already studies the scaling behavior of masked and uniform discrete diffusion and explicitly argues that uniform diffusion is a promising and underexplored regime.
> > >
> > > Even if von Rütte et al. is treated as prior public work, **the overlap is still substantially narrower than the review suggests.**
> > >
> > > We explicitly state that perplexity is informative within a diffusion family but can be misleading across families, and we support that claim with evaluations that go beyond likelihood scaling. Concretely,
> > > 1. At 1.7B parameters, Duo beats AR and MDLM on maths/reasoning, and our speed–quality analysis shows that models with worse likelihood scaling can still be preferable in practice.
> > > 1. We study the scaling laws for AR–MDLM interpolation regime with Eso-LMs, which Rutte et al lacks.
> > > 4. We making MDLM training 12% FLOP-efficient unlike Rutte et al.
> > >
> > > Lastly, **The reviewer applies an inconsistent standard:** it accepts an essentially ELBO-driven argument in Rütte et al. as sufficient to motivate uniform diffusion, **but labels our paper “unqualified” even though our paper explicitly argues that likelihood alone is insufficient** and adds speed–quality and downstream evidence precisely to address that limitation. **We respectfully ask the reviewer to evaluate both papers under the same criterion.**
> > >
> > > ---
> > >
> > > > and then makes unqualified cross-family claims such as MDLM requiring about 16x  more compute to match AR validation loss, Duo requiring about 23x more compute to match AR perplexity, and Eso-LMs requiring about32x  more compute than AR to match perplexity
> > >
> > > **This comment selectively quotes the paper and, in doing so, mischaracterizes our claim.**
> > >
> > > When the paper states that “all comparisons are conducted under matched training FLOPs,” that refers to fairness of the setup: same backbone class, tokenizer, data regime, context length, and exact compute accounting across families. It does not mean that AR NLL and diffusion NELBOs are theoretically identical objects. In fact, **the abstract and introduction explicitly state the opposite**: perplexity is informative within a diffusion family but can be misleading across families, because different diffusion families induce different likelihood bounds.
> > >
> > > The statements that MDLM requires about 16× more compute to match AR validation loss, Duo about 23× more to match AR perplexity, and Eso-LM about 32× more to match AR perplexity **are therefore numerical intersections of fitted compute-vs-validation curves under each family’s native objective**. They are not the endpoint of our argument; they are precisely why `Sec. 3.3` and `Sec. 4` add speed-quality and downstream evaluations.
> > >
> > > To remove any ambiguity, we can revise “match” to “numerically evaluate” and explicitly label diffusion perplexity as bound-based. **The reviewer quotes the matched-FLOPs sentence while omitting the paper’s explicit caveats** about cross-family comparability, which changes the meaning of the claim.
> > >
> > > ---
> > >
> > > ## **[New Experiment]** Mauve  Scores
> > > We report the best Mauve score ($\uparrow$) achieved by an alogorithm within a throughput budget.
> > >
> > > | Method \ Throughput | <200 | 200-400 | 400-600 | > 600 |
> > > |----------------|------|---------|---------|-------|
> > > | AR             | **0.85** | -       | -      | -    |
> > > | MDLM           | 0.57 | 0.12     | -      | -    |
> > > | Eso-LMs        | -    | -       | **0.31**    | 0.12  |
> > > | Duo            | 0.41 | **0.37**    | 0.29    | **0.21**   |
> > >
> > > **The takeaway remains unchanged**: AR models dominate in the low throughput range (`< 200 toks / sec`), Duo dominates in the range `[200, 400] U [> 600] toks / sec` and Eso-LMs dominate in `[400, 600] toks / sec` range. We will include the complete pareto frontier in the next revision.
> > >
> > >
> > > ## Others
> > >
> > > > Shaul et al.'s Flow Matching with General Discrete Paths: A Kinetic-Optimal Perspective already provided a general ELBO / likelihood-bound perspective for discrete paths in 2024.
> > >
> > > Shaul et al. do derive a general ELBO perspective for discrete Flow Matching. **But that does not undercut the novelty of our paper**, because we do not claim the first general ELBO for arbitrary discrete paths. We empirically show that replacing the weighted cross-entropy objective used in prior MDLM work with a plain cross-entropy objective improves MDLM’s compute efficiency by approximately `12%` under matched-FLOP scaling. That finding is independent of the prior ELBO lineage and remains novel.

---

### Official Review · Reviewer_BXXg · 2026-03-11

**Soundness:** 3
**Presentation:** 4
**Significance:** 3
**Originality:** 2
**Overall Recommendation:** 4
**Confidence:** 3

**Summary:**

This paper studies scaling properties of different discrete diffusion language model (dLLM) formulations, with a focus on comparing masked diffusion with uniform-state diffusion and interpolating diffusion methods. The authors show that masked diffusion models can be made about 12% more FLOPs-efficient by using a simple cross-entropy objective, instead of the standard diffusion objective.

**Compliance With Llm Reviewing Policy:**

Affirmed.

**Final Justification:**

My concerns have been addressed, and I will keep my score unchanged.

However, I noticed that the **references in the paper may not follow the correct template.**

**Key Questions For Authors:**

1. Do the authors expect the insights from their analysis to lead to new architectural improvements for dLLMS?

2. Do the authors have hypotheses or theoretical insights explaining why certain diffusion formulations perform better in specific scenarios? Additional discussion or analysis could help better interpret the empirical findings.

3. Although the paper studies models up to 1.7B parameters, this scale is still relatively small compared to modern large language models. Do the authors have evidence or intuition suggesting that the observed scaling trends would continue to hold for significantly larger models (e.g., 7B or larger)? It would be helpful to discuss potential limitations or expected changes when scaling further.

**Limitations:**

See weaknesses and questions above. I'm welling to adjust my score according to the authors' rebuttal.

**Strengths And Weaknesses:**

Strengths:

1. The paper challenges a widely held assumption: Masked diffusion is the dominant diffusion LM formulation. By conducting scaling studies across different diffusion families, the work provides valuable empirical insights into diffusion model design.

2. The paper shows that: Perplexity does not necessarily correlate with downstream performance Diffusion models with worse likelihood may still perform better on reasoning tasks. This observation could have important implications for evaluation practices in diffusion LMs.

3. Very well written. Easy to follow.

Weaknesses:

1. The paper mainly presents empirical observations and scaling analyses, but introduces limited methodological innovation. There is no fundamentally new modeling framework or algorithm proposed (but it is still a valuable work).

2. Despite discussing scaling behavior, the paper provides little theoretical explanation for why different diffusion formulations behave differently.

3. While 1.7B parameters is large for dLLMs, it is still small compared to modern language models. It is unclear whether the observed scaling behavior would hold at larger model sizes.

---

> ### Author Rebuttal · Authors · 2026-03-31
>
> We thank the reviewer for the thoughtful feedback and for recognizing that the paper:
> 1. **Challenges a widely held assumption** that masked diffusion is the dominant formulation for diffusion LMs,
> 2. **questions the metrics** currently used to compare various categories of diffusion LLMs in the literature, and
> 3. contributes to **improving the training efficiency of MDLM**. We address the concerns below.
>
> # Concern: Scaling to larger model sizes
>
> We agree that larger-scale experiments at 8B+ would be valuable, but they are infeasible within our resource constraints. Our study spans four paradigms (AR, MDLM, Duo, Eso-LMs), and for MDLM we evaluate two loss formulations. Scaling experiments alone required ~50,000 H100 hours. **Training four 1.7B models required 250,000 H100 hours ($1M USD on Lambda cloud computing service)**.
>
> **We estimate that Scaling each of these models at 7B scale would require** 600000 H100 Hours (**> 2 Million USD**) which
>
> We hope the community finds the experiments and evaluations in this paper, using models up to 1.7B scale, compelling enough to **reevaluate the current trend of simply scaling masked diffusion models** [1, 2], and to **reconsider not only how diffusion language models should be evaluated**, but also which directions are most promising for their future development.
>
>
> > Do the authors have evidence or intuition suggesting that the observed scaling trends would continue to hold for significantly larger models (e.g., 7B or larger)?
>
> Note that **Masked Diffusion Models (MDLM) has already been scaled at 100B parameter scale** [1, 2]. Since Uniform-state diffusion and Eso-LMs also follow a well behaved scaling law – infact **they have a similar slope to MDLM on the likelihood-vs-flops curve** (`Fig. 3`) and vary only in the offset, **we don’t expect any issues with scaling these models to larger scales**.
>
>
> [1] Nie et al., “Large Language Diffusion Models”, 2025.
>
> [2] Bie et al., “LLaDA2.0: Scaling Up Diffusion Language Models to 100B”, 2025.
>
>
> ---
>
> # Comments
>
> > Do the authors have hypotheses or theoretical insights explaining why certain diffusion formulations perform better in specific scenarios? Additional discussion or analysis could help better interpret the empirical findings.
>
>
> We thank the reviewer for this helpful suggestion. Consider Duo vs. MDLM: each denoising step has roughly the same cost in both models, and in `Figs. 6, 7` we observe that **Duo consistently produces higher-quality samples for a similar sized model across all sampling steps $T$**. We believe this is because **Duo supports self-correction**—errors made in earlier denoising steps can be revised later, unlike in MDLM as stated in [1]. As a result, **Duo is Pareto-dominant over MDLM** across throughput regimes in `Fig. 1`.
>
> **Eso-LMs** [2], an MDLM variant, **is optimized for faster inference: it supports KV caching** and **performs a forward pass of the masked tokens supposed to be denoised** instead of the full 2048-token context (in our case) at each step. This allows it to perform more denoising steps within a fixed time budget which most likely why it outperforms Duo in the `400 - 600 toks / sec` throughput range. At higher throughput (>600 tokens/s), however, **Duo regains the advantage because larger Duo models achieve better quality in fewer steps**. We will revise the discussion of `Fig. 1` to make this intuition clearer.
>
>
> [1] Sahoo et al., "The Diffusion Duality", ICML 2025
>
> [2] Sahoo et al., "Esoteric Language Models", 2025

---

> > ### Author Rebuttal · Reviewer_BXXg · 2026-04-01
> >
> > Thanks for the response. All my concerns are addressed. I'll keep my positive score unchanged. Good luck.

---

> > > ### Author Response · Authors · 2026-04-06
> > >
> > > We are grateful to the reviewer for their careful review and constructive feedback on the paper.

---

### Official Review · Reviewer_4vNt · 2026-03-12

**Soundness:** 3
**Presentation:** 4
**Significance:** 3
**Originality:** 3
**Overall Recommendation:** 4
**Confidence:** 3

**Summary:**

This paper presents a series of scaling experiments for diffusion language models (DLMs). The paper conducts comparisons among autoregressive models, masked DLMs (MDLMs), and uniform-state DLMs. The authors aim to demonstrate the favorable properties of uniform-state DLMs beyond the more commonly used MDLMs.

**Compliance With Llm Reviewing Policy:**

Affirmed.

**Final Justification:**

All my concerns in the previous review have been resolved.

**Key Questions For Authors:**

1. The authors mainly use Duo as the representative model for uniform-state DLMs. However, there are many other variants of uniform-state DLMs, such as UDLM [1], D3PM-Uniform [2], and SEDD [3]. Would the conclusions differ for these variants compared to those presented in the paper?

2. It is not clear how the validation loss is computed for different models. It would help readers better understand the scaling curves if the authors clarified whether the validation loss is computed using the same objective as the training loss (but evaluated on the validation set) for different models. To better demonstrate the scaling effect, would it be necessary to unify the validation loss across the compared models?

3. When measuring the inference speed of DLMs, what is the average context length or sampler block size used to run bidirectional attention? Will the throughput curves in Fig. 1 for DLMs be significantly affected by the block size, given that DLMs need to process the entire block at each sampling step?


[1] Schiff, Yair, et al. "Simple guidance mechanisms for discrete diffusion models."

[2] Austin, Jacob, et al. "Structured Denoising Diffusion Models in Discrete State-Spaces."

[3] Lou, Aaron, et al. "Discrete diffusion modeling by estimating the ratios of the data distribution."

**Limitations:**

See the "key questions" section. If some of them remain hard to reply, you can leave them in a limitation discussion.

**Strengths And Weaknesses:**

Strengths:

1. This paper presents new and interesting scaling experiment results for DLMs, especially including uniform-state DLMs, which should be an interesting empirical study in this community.

2. The paper elaborates on the details of how to construct scaling curves and evaluate the scaling behavior of DLMs from multiple aspects.

Weaknesses:

1. As an empirical study, the main results in the paper appear somewhat sparse. For example, in Table 1, including more baseline results and additional variants of DLMs could provide a more comprehensive comparison.

2. Some experimental details do not seem to be clearly provided in the paper.

3. The largest models used in the scaling experiments have 1.7B parameters, which are relatively small. Ideally, results for models of at least 7B parameters would better demonstrate the scaling behavior.

---

> ### Author Rebuttal · Authors · 2026-03-31
>
> We thank the reviewer for the thoughtful feedback and for **recognizing the novelty of our scaling experiments on Masked, Uniform-state and Esoteric LMs**.
>
> # A note on the novelty
>
> The core contribution of the paper: **it reveals that the metrics currently used to evaluate and compare diffusion LLMs can be fundamentally misleading**.  This matters because the community has **increasingly concentrated on scaling masked diffusion models** [1], largely due to the belief that other diffusion formulations, such as uniform-state diffusion, are inferior. **Our findings overturn that narrative.** Despite having worse perplexity, Uniform-state diffusion (Duo) outperforms MDLM on math and reasoning tasks which suggests that **the field may be prioritizing the wrong model families based on incomplete evaluation criteria**.
>
> [1] Bie et al., “LLaDA2.0: Scaling Up Diffusion Language Models to 100B”, 2025.
>
> ---
>
> We address the concerns below.
>
> # Concern: Lack of baselines
>
> >  in Table 1, including more baseline results and additional variants of DLMs
>
> In `Tab. 1`, we **retrain state-of-the-art diffusion models across all major categories**: MDLM (masked diffusion), Duo (uniform-state), and Eso-LMs (interpolating diffusion and autoregressive models). Probably, the reviewer is asking us to include other variants such as UDLM, SEDD etc which we address subsequently.
>
> ---
>
> > The authors mainly use Duo for uniform-state DLMs. However, there are many other variants of uniform-state DLMs, such as UDLM, D3PM-Uniform, and SEDD. Would the conclusions differ for these variants compared to those presented in the paper
>
> Duo’s training objective is a Rao–Blackwellized version of UDLM [2], which itself is a Rao-Blackwellized version of SEDD-Uniform [3]. Thus, all three methods are equivalent when marginalized over data and diffusion time, differing only in training variance. SEDD-Uniform has the highest variance, while Duo has the lowest; see [1, 2] for a discussion. This explains why Duo (without curriculum learning) achieves better perplexity than UDLM (Tab. 3 in [1]), which in turn outperforms SEDD-Uniform (Tabs. 2, 3 in [2]) with the exact same training setup. Furthermore, [1, 2, 3] induce the tightest possible likelihood bound over D3PM-Uniform [4]; hence, **[2, 3, 4] will exhibit worse scaling behavior compared to Duo [1]**.
>
> [1] Sahoo et al., “The diffusion duality”, ICML 2025
>
> [2] Schiff et al., “Simple Guidance Mechanisms for Discrete Diffusion Models” ICLR 2025
>
> [3] Lou et al., “Score Entropy Discrete Diffusion”, ICML 2024.
>
> [4] Austin et al., D3PM, NeurIPS 2021.
>
> ---
>
> > Ideally, results for models of at least 7B parameters would better demonstrate the scaling behavior.
>
> We agree that larger-scale experiments would be valuable, but they are infeasible within our resource constraints. Our study spans four paradigms (AR, MDLM, Duo, Eso-LMs), and for MDLM we evaluate two loss formulations. Scaling experiments alone required ~50,000 H100 hours. **Training 4 1.7B models required 250,000 H100 hours ($1M USD on cloud computing service)**.
>
> **We estimate that Scaling each of these models at 7B scale would require** 600000 H100 Hours (**> 2Million USD**) which is beyond the capacity of our lab.
>
> ---
>
> # Minor Comments
>
> > Some experimental details do not seem to be clearly provided
>
> We provide details on model architecture (`L[237–255]`), data and tokenizer (`L[257–268]`), optimizer (`L[297–300]`), model sizes (`Tab. 4`), and compute budgets (`L[303–306]`) for scaling experiments. Details for the 1.7B model are included in `L[363–387]`. If any information is missing, we are happy to clarify.
>
> ---
>
> > It is not clear how the validation loss is computed for different models
>
> We specify the negative evidence lower bound (NELBO): Duo (`L[189 - 190]`), MDLM (`L213`), and Eso-LMs (`L[249-251]`). Validation perplexity is computed on a held-out set using the respective NELBO for each method.
>
> ---
>
> > would it be necessary to unify the validation loss across the compared models?
>
> This is a valid concern. Current diffusion literature compares models using perplexity despite differing NELBO formulations, which may not reflect true likelihood ordering. **Our paper raises this issue and provides evidence that perplexity alone is insufficient**:
> (1) Duo and Eso-LMs have worse perplexity than MDLM but are Pareto superior in the speed–accuracy tradeoff (`Fig. 1`).
> (2) Duo outperforms MDLM on math and reasoning tasks despite worse perplexity (`Tab. 2`).
>
> However, this doesn't affect the scaling law for each model category.
>
> ---
>
> > When measuring the inference speed of DLMs, what is the average context length or sampler block size used to run bidirectional attention?
>
> We refrain from block-wise decoding following Eso-LMs [1] where they demonstrate that the sample quality degrade sharply as NFEs decrease. Therefore, we perform diffusion over the full context length of 2048 for Duo, MDLM, and Eso-LMs.
>
> [1] Sahoo et al., “Esoteric Language Models”, 2025

---

> > ### Author Rebuttal · Reviewer_4vNt · 2026-04-04
> >
> > All my concerns have been resolved. I will change my rating to 4.

---

> > > ### Author Response · Authors · 2026-04-06
> > >
> > > We thank the reviewer for finding our clarifications valuable and for agreeing to increase the score. We would also like to **gently remind them to update the score in the original review** to reflect this.
> > >
> > > Additionally, we are grateful for their careful review and constructive feedback on the paper.

---

### Decision · Program_Chairs · 2026-04-30

**Decision:**

Accept (regular)

**Comment:**

This paper presents the scaling behaviors of discrete diffusion language models with focus on masked diffusion, uniform-state diffusion, and interpolating diffusion methods.

### Strengths

- The paper tackles an important and timely empirical question for diffusion language models: how different diffusion families compare under compute-matched scaling and whether masked diffusion should really be treated as the default future direction.
- The experimental study is broad and useful. A central message of the paper is that it challenges perplexity-centric evaluation and argues that models with weaker likelihood scaling may nevertheless be preferable in practice.

### Weaknesses

- Although 1.7B is already expensive for this class of models, the paper still does not settle whether the observed trends persist at substantially larger scales, say 7B. Several reviewers viewed as important for the broader claim.
- The paper would benefit from clearer positioning relative to closely related recent work.
- One reviewer raised a substantive concern that some cross-family perplexity/loss statements should be phrased more carefully, because diffusion models are compared through bounds rather than directly comparable NLL quantities.
- Some useful evaluation details and additional metrics, particularly MAUVE and clearer reporting around cross-family quality/diversity tradeoffs, would strengthen the final presentation.

### Rebuttal and Remaining Concerns

During the rebuttal, the authors clarified the role of Duo relative to other uniform-state formulations, explained why larger-scale experiments were infeasible, added intuition for why different diffusion families may behave differently, clarified dataset and validation-loss details, and provided additional MAUVE results. Three reviewers explicitly indicated that their concerns were addressed and ended at weak accept.

The main remaining concerns come from one reviewer who maintained a reject score:

- clearer positioning with respect to closely related recent work;
- more careful wording around cross-family perplexity comparisons when diffusion models are evaluated via bounds rather than directly comparable likelihoods; and
- inclusion of additional evaluation material such as MAUVE in the revision.

The paper is on the borderline with core empirical contributions valuable to the community. I would encourage the authors to revise carefully to improve the paper's presentation and make the evaluation more convincing.